# Random sequences rapidly evolve into de novo promoters

Avihu H. Yona[1,2], Eric J. Alm[2] & Jeff Gore [1]

How new functions arise de novo is a fundamental question in evolution. We studied de novo evolution of promoters in *Escherichia coli* by replacing the lac promoter with various random sequences of the same size (~100 bp) and evolving the cells in the presence of lactose. We find that ~60% of random sequences can evolve expression comparable to the wild-type with only one mutation, and that ~10% of random sequences can serve as active promoters even without evolution. Such a short mutational distance between random sequences and active promoters may improve the evolvability, yet may also lead to accidental promoters inside genes that interfere with normal expression. Indeed, our bioinformatic analyses indicate that *E. coli* was under selection to reduce accidental promoters inside genes by avoiding promoter-like sequences. We suggest that a low threshold for functionality balanced by selection against undesired targets can increase the evolvability by making new beneficial features more accessible.

[1] Physics of Living Systems, Department of Physics, Massachusetts Institute of Technology, Cambridge, MA 02139, USA. [2] Department of Biological Engineering, Massachusetts Institute of Technology, Cambridge, MA 02139, USA. Correspondence and requests for materials should be addressed to A.H.Y. (email: avihu.yona@gmail.com) or to J.G. (email: gore@mit.edu)

D e novo evolution of complex traits may require a combination of genetic changes before a beneficial function can be obtained[1]. In such cases, the evolutionary path is not trivial, as a negligible selective advantage of first mutations may prevent them from spreading in the population and further acquire the other needed mutations. The possibility of acquiring multiple desired mutations simultaneously (rather than serially) has very low probability, especially in asexual populations, such as bacteria, that are unable to combine mutations that were acquired in different individuals[2].

The *Escherichia coli* promoter represents a complex sequence feature as it consists of different elements that act together to transcribe a gene. The RNA polymerase requires particular sequence elements for binding, and additional features, such as transcription factors and small ligands can further affect its activity. The canonical *E. coli* promoter ($\sigma^{70}$) is recognized by consensus sequence elements, the two principal ones being the −10 element TATAAT and the −35 element TTGACA, which are separated by a spacer with an optimal length of 17 bases. Additional sequence elements, such as the extended −10 (TGn) and the UP elements, can be recognized as well, and they act together for the promoter to be recognized by the RNA polymerase[3]. These features make promoter evolution a promising avenue to consider how complex features can evolve.

The extensive study of promoters by genomic analysis[4–6], experimental protein–DNA interactions[7–9], and promoter libraries has mostly revolved around the evolutionary-refined promoters, i.e., long-standing wild-type promoters and their derivatives[10–12]. Yet, the first evolutionary step of a new promoter emerging from scratch are less understood, for example, when cells need to activate new[13,14] or inactive[15] genes. Studies following how inactive genes evolve the expression have demonstrated that an existing promoter is often copied upstream to the gene whose expression is needed. This typically occurs via genomic rearrangements or transposable elements that contain active promoters[16–28]. Activating the genes by copying the existing promoters suggests that de novo promoters may not be very accessible evolutionarily. Preexisting similarities often occur when the inactive gene comes from another bacteria with similar promoter motifs[29], or when the inactive gene is near a native intergenic region that normally contains multiple overlapping promoter elements[30,31]. Copying the existing promoters is therefore prevalent in evolution presumably because otherwise multiple mutations are required and they take much longer time to be obtained.

To systematically study the evolution of de novo promoters, one should start from non-functional sequences. Random sequences, i.e., sequences composed of A, C, G, and T in equal probabilities contain no information and thus represent the non-active sequence space without biases. Using purely random sequences as a starting point for promoter evolution is especially suitable for genomes with ~50% GC content, such as the *E. coli* genome, which is 50.8% GC. For such genomes, random sequences can serve as a null model when testing for functionality without introducing biases or confounding factors due to deviating from the natural GC content of the studied genome.

The number of mutations needed in order to change a random sequence into a functional promoter is not clear. Especially in experimental and quantitative terms, the question is how many mutations does one need in order to make a functional promoter, starting from a random sequence of a specific length? This question can be addressed directly by experimental evolution. We evolved parallel populations, each starting with a different random sequence, which replaced the whole intergenic region from the beginning of the coding sequence and up to the terminator of neighboring gene upstream. Following these, the evolving populations highlighted that new promoters can often emerge directly by mutations, and not necessarily by genome rearrangements that copy an existing promoter. Substantial promoter activity can typically be achieved by a single mutation in a 100-base sequence, and can be further increased in a stepwise manner by additional mutations that improve similarity to canonical promoter elements. We therefore find a remarkable flexibility in the transcription network on the one hand, and a tradeoff of low specificity on the other hand, with interesting implications for the design principles of genome evolution.

## Results

**Replacing the wild-type lac promoter with random sequences**. To create an ecological scenario that can test how bacteria evolve de novo promoters, we sought a beneficial gene in the genome that is not yet expressed, similarly to what might occur when a gene is inactive or transferred horizontally without a functional promoter[32]. To this end, we modified the lac operon in *E. coli* by replacing the native promoter with random sequences. It is important to note that this work is not focused on the lac promoter or operon, as the lac promoter has been deleted, and we merely use the lac metabolic genes for their ability to confer a fitness advantage upon expression in the presence of lactose. Accordingly, we modified the lac operon so that only the lac metabolic genes (*lacZYA*) remain intact (including their 5′UTR); we deleted the lac repressor (*lacI*) and eliminated the lac promoter by deleting the entire intergenic region upstream to the lac genes and replaced it with a variety of non-functional sequences. To broadly represent the non-functional sequence space, we used random sequences (generated by a computer) with equal probabilities for all four bases (Methods).

**Typical ~100 bp random sequence disables lactose utilization**. The random sequences that replaced the WT lac prompter were 103 bases long, which is the typical length for an intergenic region in *E. coli* (the median intergenic region in *E. coli* is 134 bases[33]). It is also the exact same length as the deleted intergenic region that originally harbored the WT lac promoter. In addition, the lactose permease (*lacY*) was fluorescently labeled with YFP[34] for future quantification of expression. To avoid the possible artifacts associated with plasmids (Supplementary Note 1), all modifications were made on the *E. coli* chromosome, so the engineered strains had a single copy of the metabolic genes needed for lactose utilization, yet without a functional promoter (Fig. 1a). We began by building three such strains, each one carrying a different random sequence upstream to the lac genes (termed RandSequence1, 2, and 3). We observed that none of these strains could utilize or grow on lactose because they could not express the lac genes (Supplementary Fig. 1). This experimental observation was therefore consistent with the expectation that a random sequence is unlikely to be a functional promoter.

**Evolving random sequences on a mix of glycerol and lactose**. To evolve the de novo expression of the lac genes, we applied selection for the ability to utilize lactose. Therefore, our criterion of whether the expression is on or off was not by setting an arbitrary threshold, but rather by a functional readout—the ability to grow on lactose as a sole carbon source. We started evolution with RandSequence1, 2, and 3, each strain in four population replicates; as controls, we also evolved a strain in which the WT lac promoter was not replaced (termed WTpromoter), and another strain in which the entire lac operon (promoter and genes) was deleted (termed ΔLacOperon). Before the evolution experiment, only the WTpromoter strain could utilize lactose (Supplementary Fig. 1). Therefore, to facilitate growth to

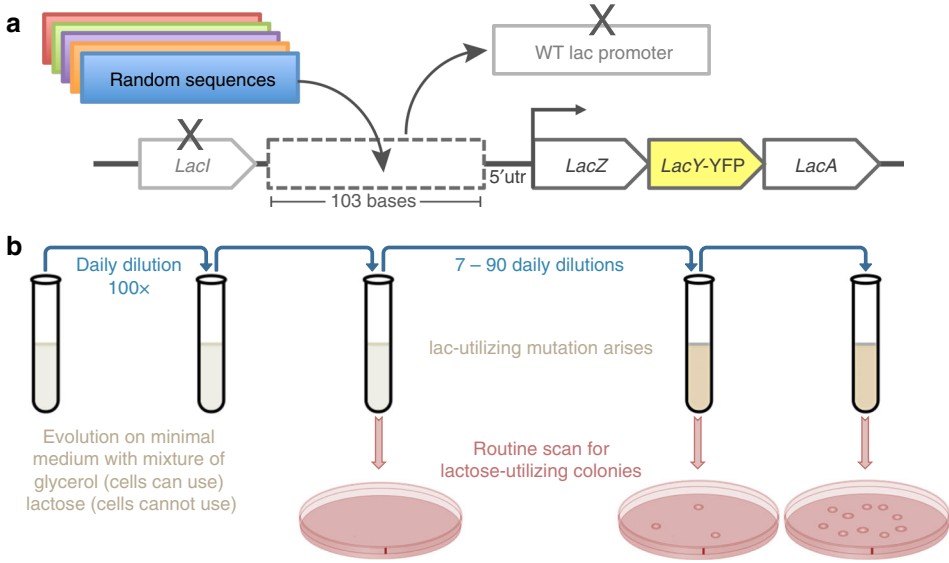

**Fig. 1** Experimental setup for evolving promoters from random sequences. **a** We modified the chromosomal copy of the lac operon by replacing the intergenic region that harbors the WT lac promoter with a random sequence of the same length (103 bases) that abolished the cells' ability to utilize lactose. In addition, the *lacY* was tagged with YFP and the lac repressor (*lacI*) was deleted. **b** Cells were evolved by serial dilution in minimal medium containing both glycerol (0.05%), which the cells can utilize, and lactose (0.2%), which the cells cannot utilize unless they evolve de novo expression of the lac genes. During evolution, samples were routinely plated on minimal medium plates with lactose as a sole carbon source, for the isolation of lactose-utilizing mutants

low population sizes, the evolution medium contained glycerol (0.05%) that the cells can utilize and lactose (0.2%) that the cells can only exploit if they express the lac genes.

**Lactose-utilizing mutants isolated on lactose only plates**. To isolate lactose-utilizing mutants, we routinely plated the samples from the evolving populations on plates with lactose as the sole carbon source (M9+Lac) (Fig. 1b). Remarkably, within 1–2 weeks of evolution (less than 100 generations), all populations acquired lactose-utilizing abilities, except for the ΔLacOperon population. These lab evolution results therefore argue that the populations carrying random sequences, instead of promoters, can rapidly evolve the de novo expression. Next, we addressed the question of whether the solutions found during evolution were mutations in the random sequences or simply copying of existing promoters from elsewhere in the genome.

**Minimal mutations turn random sequences into promoters**. To determine the molecular nature of the evolutionary adaptation, we sequenced the region upstream to the lac genes (from the beginning of the lac genes through the random sequence and up to the neighboring gene upstream). Within each of the evolved random sequences, a single mutation was found to confer the ability to utilize lactose. Continued evolution yielded additional mutations within the random sequences that further increased expression from the emerging promoters. The different replicates acquired the same mutations, yet sometimes in a different order (Supplementary Data 1). Each mutation was inserted back into its relevant ancestral strain, thus confirming that the evolved ability to utilize lactose is due to the observed mutations.

**Evolved promoters reach the expression level of the WT**. Next, we assessed the levels of the evolved expression by YFP measurements (thanks to the LacY-YFP labeling); we found that the evolved promoters produce an expression that was comparable to a fully induced WT lac promoter (Fig. 2a). This experimental evolution demonstrates how non-functional sequences can

rapidly become active promoters, in a stepwise manner, by acquiring successive mutations that gradually increase expression. Next, we aimed to determine the mechanism by which these mutations induced de novo expression from a random sequence.

**Evolved mutations mimic canonical promoter motifs**. The sequence context of the mutations that emerged in the random sequences suggests that the de novo expression has evolved by increasing similarities to the consensus sequence of the canonical promoter motifs[35]. Each of the five evolved mutations that were found in Randseq1, 2, and 3 increased the similarity to either the TATAAT or the TTGACA consensus sequences. In Randseq1, a single base substitution created an almost perfect −10 motif and a consecutive mutation further increased the expression by improving the −35 element. A similar scenario was observed in Randseq2, yet in the reverse order, as the first mutation created a −35 element and the later mutation further increased expression by improving the −10 element (Fig. 2b). In Randseq3, however, no successive mutations were observed after the first mutation that induced expression by creating a perfect TATAAT motif. The evolved mutation in Randseq3 occurred alongside an extended −10 motif[36] that enabled the expression even without a proper −35 element. Therefore, unlike Randseq1 and 2, in Randseq3 no putative −35 element could be found in a tolerable spacing from the −10 element.

To validate that the evolved mutations indeed induced expression by creating a canonical promoter, we demonstrated loss of promoter function upon mutating the most essential position within the evolved canonical promoter—the adenine in position −11. Changing the −11 adenine to guanine completely abolished the expression of RandSeq1 and RandSeq3. Yet, in the case of RandSeq2 the evolved promoter overlaps with another predicted promoter (see Supplementary Data 1); therefore we only mutated the −11 adenine of the evolved promoter. In this mutant, low expression was still detected yet expression dropped by 3.2-fold compared to the evolved RandSeq2. The loss of expression in these three mutated sequences is therefore

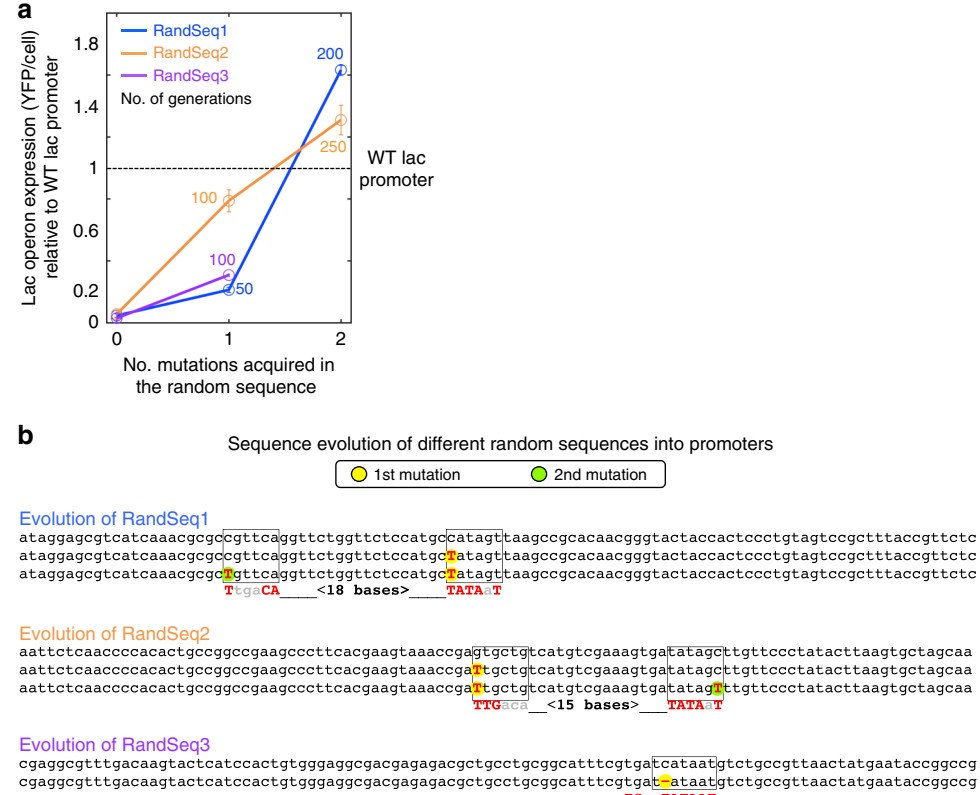

**Fig. 2** From a random sequence to an active promoter by stepwise mutations (example from three random sequences). **a** Evolved expression levels of the lac genes are plotted for three strains that carry different random sequences (blue, orange, and purple) as a function of the number of acquired mutations. Expression level of 1 is defined as the expression measured from the WT lac promoter, and 0 is defined as the background read of the control strain, ΔLacOperon (in which the lac operon is deleted and no YFP gene was integrated). After the accumulation of mutations, de novo expression is observed (as well as the ability to utilize lactose). The number of generations is indicated near each mutation. Mutations shown were verified by reinsertion into their non-evolved ancestors. Error bars represent s.d. of YFP measurements of three biological replicates. **b** Sequences of the evolving promoters. For each strain, the top sequence is the random sequence before evolution, second, and third lines are the random sequence with the evolved mutations (first and second mutations, respectively). Increasing similarity to the canonical E. coli promoter motifs can be observed by different mutations. For each evolving promoter, the canonical promoter is shown as the bottom line where capital bases indicate a match. For RandSeq3, the evolved promoter is an extended −10 promoter, thus the −35 motif is not indicated

consistent with our interpretation that the evolved solutions had indeed created the canonical promoter motifs.

The lab evolution results from RandSeq1, 2, and 3 indicate that de novo promoters are highly accessible evolutionarily, as a single mutation created a promoter motif that enabled growth on lactose, suggesting that a sequence space of ~100 bases might be sufficient for evolution to find an active promoter with one mutational step.

**~10% of random 103 bp serve as promoters without evolution.** The important step of evolving random sequences into promoters was the first mutation whose effect was sufficient to enable the growth on lactose by turning on the expression. Therefore, we predicted that, if indeed, a single mutation in a 103-base random sequence is often sufficient to generate an active promoter, there might be a small portion of random sequences that are already active without the need of any mutation. This prediction is also consistent with previous studies that utilized the selection to isolate the fraction of active sequences out of a random pool of sequences on plasmids, without prior evolution[37,38]. Indeed, when we expanded our collection to 40 strains, each carrying a different random sequence (RandSeq1 to 40), we identified four strains (10%) that could form colonies on M9+Lac plates before evolution and without acquiring any mutation in their random sequences. We scanned the random sequences of these already-

active strains (RandSeq7, 12, 30, 34) and found regions with high similarity to the canonical promoter consensus sequences, equivalent to the similarities caused by the mutations mentioned earlier for RandSeq1, 2, and 3 (Supplementary Fig. 2). Given that a single mutation might be sufficient to turn expression on, we proceeded with the strains that could not grow on lactose, by putting them under selection for lactose utilization both by the above mentioned daily-dilution routine (in M9+GlyLac) and by directly screening for mutants that can form colonies on M9+Lac plates (Methods).

**~60% of random 103 bp need one mutation to act as promoters.** Overall, the evolving expression of the lac operon by selection for lactose utilization was successful for all but two of the random-sequence strains (38/40). Analysis of all forty strains and their lac operon-activating mutations showed that: 10 ± 5% were already active without any mutation (4/40), 57.5 ± 8% found mutations within the 103 bases of the random sequence (23/40), 12.5 ± 5% found mutations in the intergenic region just upstream to the random sequence (5/40) and 15 ± 6% utilized genomic rearrangements that relocated an existing promoter of genes found upstream to the lac genes (6/40) (Fig. 3a and Supplementary Note 2). To confirm that the transcriptional read-through from the selection gene upstream did not facilitate the emergence of the de novo promoters, six strains were made in a

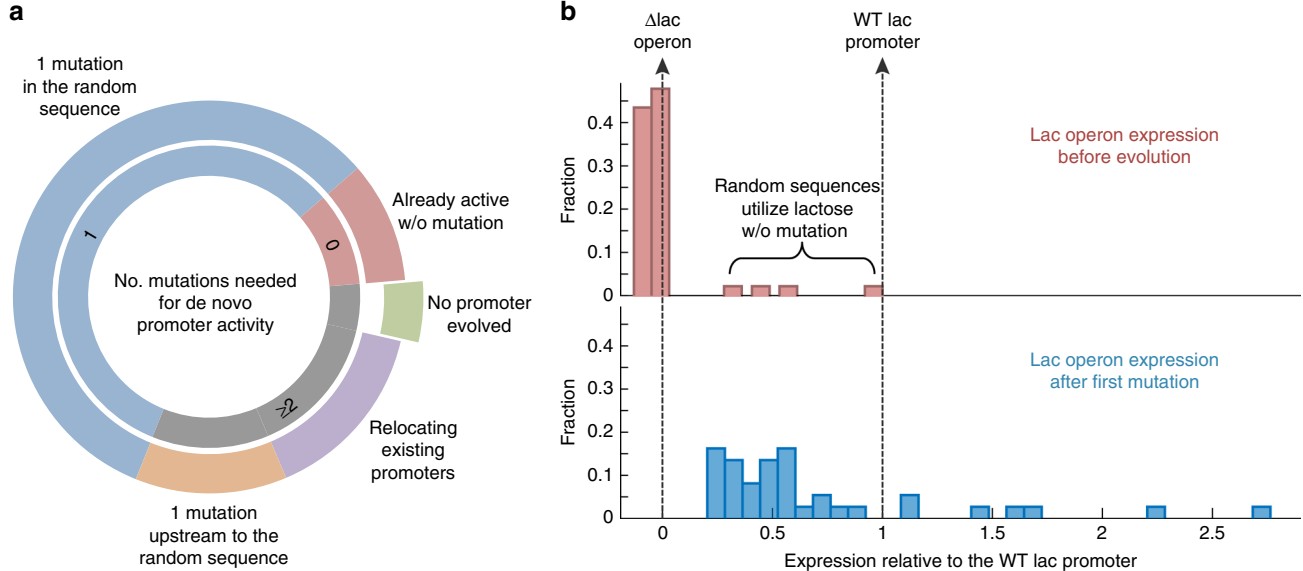

**Fig. 3** A typical random sequence of ~100 bases requires only one mutation to evolve into a promoter. **a** A summary of 40 different random sequences and the different type (outer circle) and number (inner circle) of mutations needed to transform a random sequence to a functional promoter. Zero represents random sequences that were active already before any adaptation and ≥2 represent random sequences that require two or more mutation, in which we include all the library strains that could not evolve the expression via mutations in the random sequences. ~10% of random sequences require no mutation for expression of the lac genes (red segment). For 57.5% of random sequences, a single mutation found within the random sequence enabled expression of the lac genes (similar to RandSeq 1, 2, and 3 shown earlier) (blue segment). Other strains either relocated an existing promoter from another locus in the genome to be upstream to the lac promoter (15%, purple) or found point mutations in the intergenic region upstream to the random sequence (12.5%, orange). **b** Expression of the lac genes before evolution and after the first mutation that was associated with the ability to utilize lactose (upper and lower panel, respectively). YFP reads normalized to OD600 are shown. Expression level of 1 is defined as the expression measured from the WT lac promoter (right vertical dashed line), and 0 is defined as the background read of the control strain ΔLacOperon in which the lac operon is deleted and no YFP gene was integrated (left vertical dashed line). The ~10% of random sequences that conferred the ability to utilize lactose even before evolution is found to have significant expression of the lac genes (upper panel)

marker-free manner (Methods) and showed that their ability to evolve the de novo promoters is similar to the rest of the strains. A typical random sequence of ~100 bases is therefore not an active promoter, but is often only one-point mutation away from being an active promoter.

**The first mutation typically yields ~50% of the WT activity**. YFP measurements indicated that all strains evolved substantial expression of the lac genes after acquiring the activating mutations (Fig. 3b). In particular, the strains that evolved by mutations in their random sequence exhibit a median expression equivalent to ~50% of the expression observed from a fully induced WT lac promoter (which includes a CRP transcription activator). The promoters that we evolve from random sequences therefore display significant levels of expression, and are not weak "leaky" promoters. Nonetheless, continued evolution would likely lead to increased expression (as in Fig. 2).

**Mutations increased similarity to the −10 and −35 motifs**. The vast majority of mutations that were found in the random sequences can be ascribed to increasing similarities to the two main promoter consensus sequences, the −10 and −35. Although some promoters had preexisting promoter motifs other than the −10 and −35, none of the mutations we found actually created or strengthened such motifs, like the UP element or the TGn motif (extended −10). For details on all mutations, their verifications and different outcomes between replicates see Supplementary Data 1.

A significant fraction of the strains in our evolved library (~2/3) showed parallel evolution, i.e., the same activating mutations occurred in the different population replicates. This indicates that

a ~100-base random sequence typically does not include multiple segments that can evolve into a promoter by a single mutation. We also saw parallel evolution in the random sequences that evolved by multiple stepwise mutations, yet the mutations sometimes occurred in a different order (like in RandSeq2, see Supplementary Data 1). Interestingly, these stepwise mutations show no signs of epistasis, as their contribution to the expression level is largely additive.

**Computational assessment of promoter sequence accessibility**. Our evolution experiment showed that a single mutation could often produce expression levels similar in magnitude to the expression level produced by the WT lac promoter. To get a numerical perspective on these findings, we assessed the mutational distance that separates random sequences from the canonical promoter of *E. coli*. To this end, we computationally created 30,000 random sequences (the same way the experimental RandSeq1 to 40 were generated) and ran a template of the canonical promoter against their sequences, using a sliding window. Since the importance of each base for promoter activity differs considerably, we weighted each base according to the position-specific matrix of the *E. coli* canonical promoter (Methods). In the same way, we also obtained scores for the 556 constitutive promoters[39] of *E. coli* and set their median score as the benchmark that qualifies as a promoter (Supplementary Note 3). The results from this analysis showed that the fraction of random sequences that qualify as promoters and the fractions that are one mutation away from a promoter coincide with the fractions observed in our experimental library. Similar fractions were also observed when the benchmark for a promoter was set to

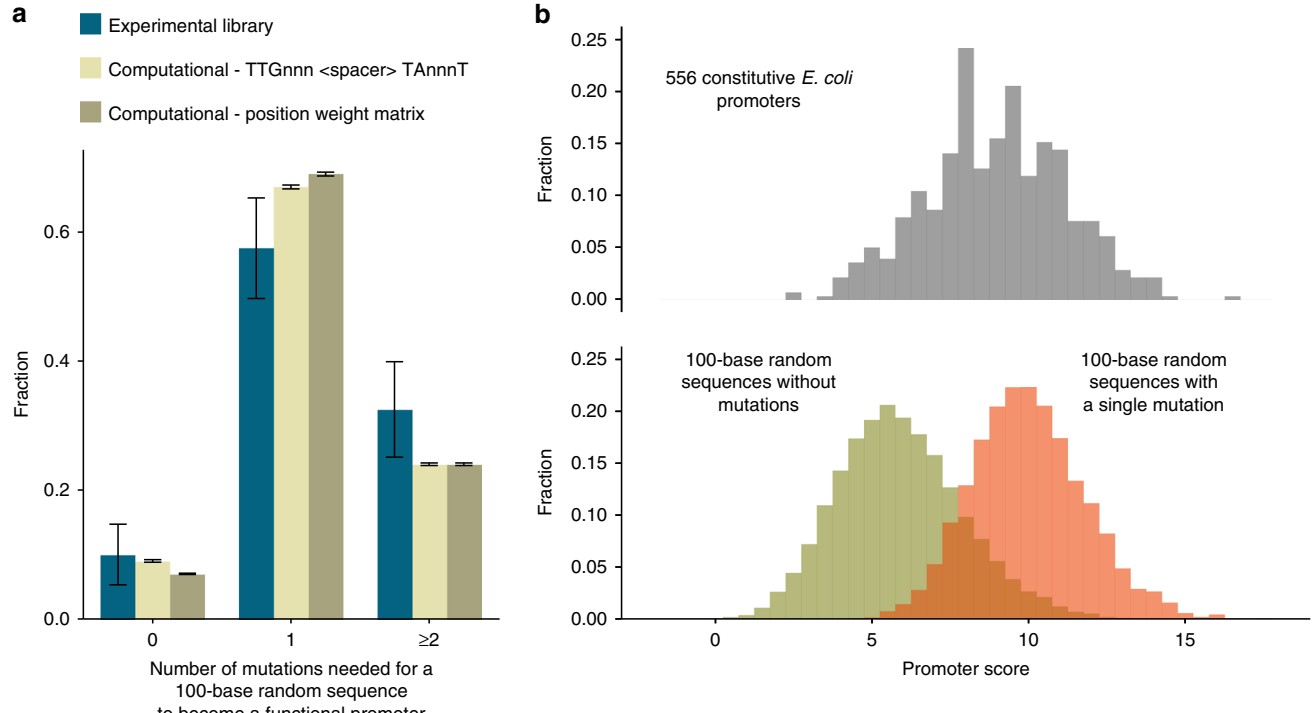

**Fig. 4** The mutational distance from a 100-base random sequence to a promoter. **a** The number of mutations needed to transform a random sequence to a functional promoter. The experimental results from the evolution library ($n = 40$) are shown in blue, zero represents random sequences that were active already before any adaptation and $\geq 2$ represent random sequences that require two or more mutation in which we include all the library strains that could not evolve expression via mutations in the random sequences. Computationally generated sequences ($n = 30,000$) were scanned for their resemblance to a canonical promoter and were mutated in silico until they pass our criterion for a promoter. We used two different criteria: the ability to capture the most important bases in the $-10$ and $-35$ promoter motifs TTGnnn and TAnnnT, with a valid spacer (light green), and the ability to score as the median score of E. coli WT constitutive promoters, according to a position-specific weight matrix score (olive). Both criteria yield similar results to those of our experimental library (error bars represent s.d.). **b** Comparing promoter scores according to a position-specific weight matrix. Upper panel shows a histogram of scores for E. coli constitutive promoters ($n = 556$). Lower panel shows a histogram of scores for random generated sequences ($n = 30,000$), before in silico evolution (olive), and after the first mutation selected (orange). The overlap between the scores of the constitutive promoters to those of the random sequences (before evolution) suggests for our experimental observation of the fraction of random sequences that are already active promoters. The overlap between the scores of the constitutive promoters to those of mutated random sequences strengthen our experimental result that a random sequence of ~100 bases is typically one mutation away from functioning as a promoter. Data from these histograms were the base for the data shown in sub- figure A (olive bars)

capture the core bases of the canonical motifs (TTGnnn and TAnnnT, where 'n' represent any base) (Fig. 4).

**The costs of highly accessible promoters**. The short mutational distance from random sequences to active promoters may act as a double-edged sword. On the one hand, the ability to rapidly "turn on" expression can provide plasticity and high evolvability to the transcriptional network. On the other hand, this ability may also impose substantial costs, as such a promiscuous transcription machinery is prone to expressing unnecessary gene fragments[40]. Spurious promoters may not only be wasteful, they can be harmful too. When promoters occur inside a coding region they recruit RNA polymerases that can interfere with the polymerases recruited by the normal promoter of the gene, and thus compromise its proper function[41–44]. We term such promoters as accidental promoters and their effect as accidental expression (not to be confused with expression noise). Our experiments indicate that ~10% of 100-base sequences can function as an active promoter, meaning that a typical ~1 kb gene might naturally contain an accidental promoter inside its coding sequence, both in the "sense" and the "anti-sense" direction. Therefore, we looked for strategies that E. coli might have taken to minimize such accidental expression.

**Indications for selection against accidental promoters**. To test whether the E. coli genome avoids accidental promoters from occurring inside genes, we computationally scanned the WT genome and identified putative promoters within the coding region. In order to assess whether the WT genome of E. coli has minimized accidental promoters we also scanned a thousand alternative versions of the E. coli genome (generated in silico) and compared their accidental promoters to those of the WT genome. We use the term "accidental expression" to describe putative expression from promoters that were predicted by sequence analysis of the coding region.

We generated a thousand alternative genomic versions of E. coli by recoding the coding region in silico (the coding region makes 88% of the E. coli genome). Since each amino acid can be encoded by multiple synonymous codons, there are many alternative ways to encode each of the genes in the genome. We therefore computed a thousand alternative versions of the E. coli genome by recoding each of its genes, while preserving the WT amino-acid sequence and the overall codon bias (Methods).

Our main method to identify promoters inside genes was based on BPROM[45,46], an available software for promoter detection in E. coli that takes into account not only the $-10$ and $-35$ elements but also other factors that affect transcription level, like the TGn element and transaction factor binding sites, as well as the

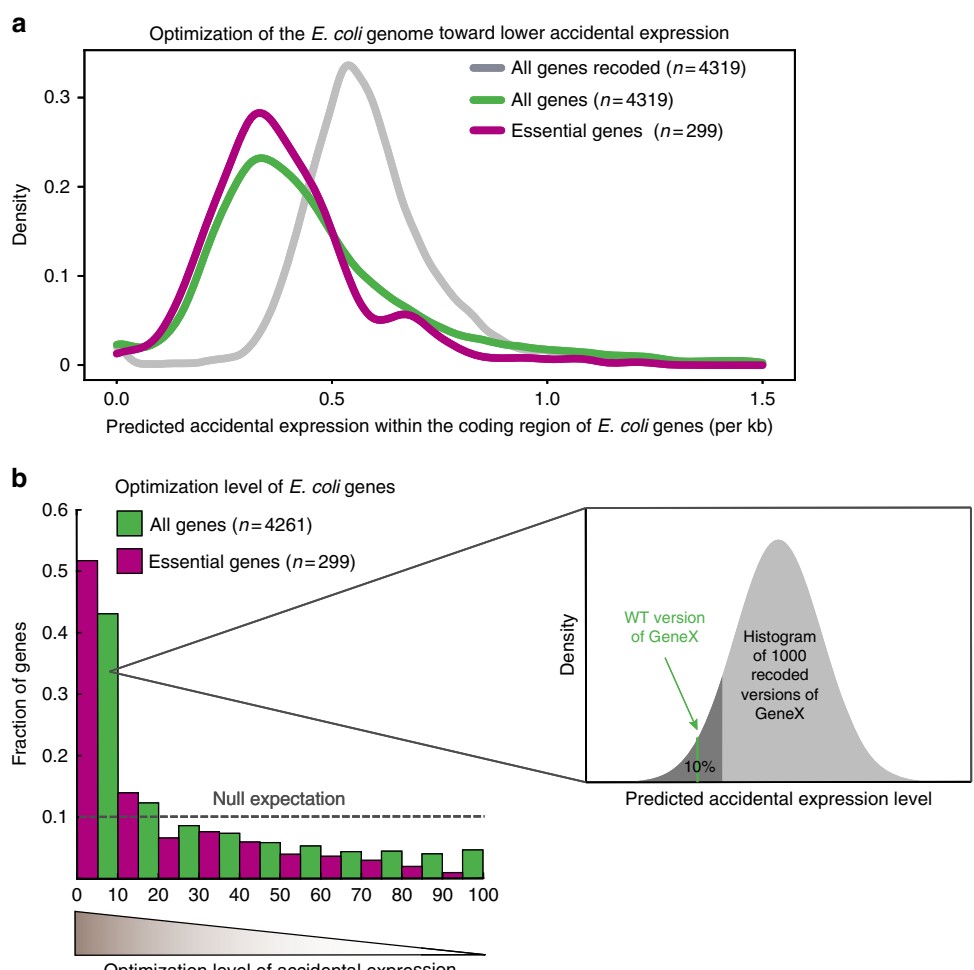

**Fig. 5** Selection against the occurrence of random promoters in the coding region of genes. We evaluated promoters that accidentally occur across the genome by searching for promoter motifs in the coding region of *E. coli*. As a reference we did the same evaluation for 1000 alternative versions of the *E. coli* coding region by recoding each gene with synonymous codons while preserving the amino-acid sequence and the codon bias. **a** Density plots of accidental expression in the coding sequences of *E. coli* genes. Distribution of a thousand recoded versions of *E. coli* coding region are shown in gray (the value that represent each gene is the median of its 1000 shuffled versions), the accidental expression of the WT *E. coli* genes is shown in green, and for the subset of essential genes in magenta. The WT version of the genome shows lower rates of predicted accidental promoters, indicating genome-wide minimization of accidental expression. This minimization is further emphasized for the essential genes. **b** For each WT gene and its 1000 recoded versions a score for accidental expression was calculated. The WT gene was then ranked in the distribution of its 1000 recoded versions (see inset illustration). Ranking values are divided into deciles, for all WT genes (green), and for the subset of essential genes (magenta) demonstrating that ~40% of WT genes and more than 50% of essential genes are ranked at the most optimized decile. Dashed line shows expected histogram if WT genes had similar values to their recoded versions

spacing between the elements (Methods and Supplementary Note 4). In addition, we performed a "six-mer" analysis—a straightforward unbiased analysis in which we count the number of occurrences for all possible six-mer motifs across the genome. Comparing the differences in six-mer occurrences between the WT genome to the recoded ones revealed which motifs are overrepresented and which are underrepresented in the WT *E. coli* genome (Methods).

We used both of these methods to compare predicted accidental promoters of the WT *E. coli* with its thousand recoded versions and found that both methods predict significantly less than expected accidental expression in the WT *E. coli*. In other words, among all the possible genomic versions that can encode the *E. coli* proteome, the WT version seems to be one of the lowest in terms of accidental expression (Fig. 5a, Supplementary Fig. 3 and Supplementary Table 1). The *E. coli* genome has therefore likely been under selection to decrease accidental expression within the coding region of genes.

**Accidental expression is not evenly minimized across genes.** A general depletion of binding sites across the genome was previously proposed to be selected in order to prevent the transcription machinery from being drained[47]. Therefore, the suggested underlying evolutionary process started with a genome that contains many spurious promoters, each providing a very low advantage when deactivated by mutation[47,48]. Nonetheless, we hypothesized that selection for minimization of accidental expression may have been driven by a more specific need to avoid interference in genes whose expression is important for the cell's fitness, rather than uniformly across the genome. To this end, we assessed the optimization level of each gene separately, by comparing the accidental expression score of each WT gene to the scores of its thousand alternative recoded versions.

Remarkably, we found that ~40% of WT genes had accidental expression as low as the lowest decile of their recoded versions. Our data indicated that some *E. coli* genes minimize the accidental expression more than others. Essential genes, for

example, exhibited an even stronger signal of optimization, compared to the general signal obtained for all genes together ($P < 0.001$, KS test) (Fig. 5b). Essential genes are presumably under stronger selective pressure to mitigate interference[49,50] caused by collisions (head-on and head-to-tail) between the RNA polymerases recruited by the accidental promoter to those recruited by their normal promoter[43,44,51]. We obtained similar results also when we used an alternative recoding method, in which we just shuffled the codons of the WT version of each gene; again indicating for selection on *E. coli* genes to minimize the occurrences of accidental promoters within them (Supplementary Fig. 4, Supplementary Note 5 and Methods).

**Interfering promoters might be used to restrain toxin genes.** Another indication that selection against accidental promoters occurs at the gene level was observed when we analyzed accidental expression in toxin/antitoxin[52] gene couples. We observed that for 80% of these couples, the toxin genes had higher accidental expression, compared to their antitoxin counterparts ($P = 0.016$, binomial) (Supplementary Fig. 5, Supplementary Note 6 and 7). Interestingly, when we split the accidental expression score into its "sense" (same strand as the gene) and "antisense" (opposite strand) components, we observed that toxins had a much stronger accidental expression in their antisense direction, compared to their sense direction ($P = 0.059$, KS test); unlike the antitoxins in which no significant difference was observed ($P = 0.541$, KS test), similar to the rest of the genome (Supplementary Fig. 6). This leads us to speculate that *E. coli* might have utilized the accidental expression as a means to restrain gene expression[53,54] of specific genes, presumably by allowing head-to-head collisions of RNA polymerases[43,44,51].

Overall, our computational analyses suggest that the promiscuous nature of the promoter recognition machinery in *E. coli* may have driven a selection pressure to deactivate instances of accidental promoters when they caused significant reduction in fitness.

## Discussion

Our study suggests that the sequence recognition of the transcription machinery is rather permissive and not restrictive[55], to the extent that the majority of non-specific sequences are on the verge of operating as active promoters. This proximity of non-functional sequences to active promoters might explain part of the pervasive transcription seen in unexpected locations in the bacterial genomes[40].

The activating mutations found in our experiments suggest that new promoters largely emerge by creating the canonical RNAP binding sites (TATAAT and TTGACA). It is not obvious why the alternative strategy of evolving binding sites for transcription factors has not been widely observed in our library. However, in the two cases, RandSeq29 and RandSeq40, we observed that predicted promoters in the random sequences were not active before evolution until the activated mutations occurred downstream to their TATAAT element, but did not create a promoter motif (see Supplementary Data 1). In these two promoters, it is possible that the evolved mutations (which occurred repeatedly) either strengthen a transcription-activator binding site or reduce the affinity of a repressor that blocked the expression from the predicted promoter upstream. Especially since, in the both cases, two different mutations evolved next to one another, which might indicate targeting a single transcription factor binding site.

Despite generating expression levels similar to the WT lac promoter, the promoters evolved in our library are of very low complexity, as most of the activating mutations involved no additional factors but the two basic promoter motifs. Although

the evolved promoters likely have no regulation, we hypothesize that such crude promoters might play an important role in the evolution of the transcriptional network, as newly activated genes do not necessarily require the regulated/induced expression in order to confer significant advantage. Furthermore, such stripped down promoters can serve as an evolutionary stepping-stone until regulation evolves, perhaps also by stepwise point mutations.

Bacterial cells can decrease accidental expression by the coiling of their chromosome, which hinders RNA polymerase from interacting with promoters, for example, histone-like proteins[56,57]. We suggest that accidental expression can also be avoided by selection against promoter-like sequences at the gene-level. Specifically, parts of a coding sequence that resemble promoter motifs can be deactivated in genes whose expression might be sensitive to interference with the internal expression (like essential genes). Promoter-like sequences in the coding region can be avoided by using alternative codons or by changing to a different amino acid, as genes can often tolerate changes in their amino sequences without a substantial effect on their function, especially if the change is to an amino acid with similar properties. Such a use of alternative codons and amino-acid substitutions might actually be one of the constraints that have shaped the codon preferences observed in the genome[58].

Our main findings may be relevant to other organisms and to other DNA/RNA binding proteins, such as transcription factors. The mutational distance between random sequences to any sequence feature should be considered for the possible "accidental recognition" and for the ability of non-functional sequences to mutate into functional ones. Therefore, the implications of this study may also prove useful to synthetic biologic designs, as one needs to be aware that non-specific sequences might not always be non-functional, as assumed. Moreover, spacer sequences can actually be properly designed to have lower probability for accidental functionality, for example, a spacer that has particularly low chances of acting as a promoter (or ribosome binding site, or any other sequence feature[58]).

Tuning the promoter recognition machinery to such a low specificity so that one mutation is often sufficient to induce substantial expression is crucial for the ability to evolve de novo promoters. If two or more mutations were needed in order to create a promoter, cells would face a much greater fitness-landscape barrier that would drastically reduce their ability to evolve the promoters de novo. In such cases, cells are likely to copy the existing promoters via genomic rearrangements. Furthermore, if a single mutation would only have a minor effect on expression, i.e., creating a very weak promoter, promoters with WT-like activity would take longer to evolve in response to new ecological challenges.

The rapid rate at which new adaptive traits appear in nature is not always anticipated, and the mechanisms underlying this rapid pace are not always clear. As part of the effort to reveal such mechanisms[59], our study suggests that the transcription machinery was tuned to be "probably approximately correct"[60] as means to rapidly evolve de novo promoters. Setting a low threshold for functionality, on one hand, while eliminating the undesired off-target instances on the other hand, makes a system where new beneficial traits are highly accessible without enduring the low-specificity tradeoffs. Further work will be necessary to determine whether and how similar principles affect the regulatory network and protein–protein interaction network in bacteria as well as in higher organisms.

## Methods

**Strains**. Strains were constructed using the Lambda-Red system[61], including integration of random sequences as promoters by using chloramphenicol resistance selection gene. Yet, for the strains with RandSeq9, 12, 15, 17, 18, 23, integration was done by the Lambda-Red-CRISPR/Cas9 system without introducing a selection marker, in order to exclude potential transcriptional read-through due to the an upstream selection gene. The ancestral strain for all 40 random sequence strains, as well as for the control strain ΔLacOperon was SX700[34] (also used as the control

strain termed WTpromoter) in which the *lacY* was tagged with YFP. In addition, the *mutS* gene was deleted (by gentamycin resistance gene) to achieve higher yield in chromosomal integration using the lambda-red system[62] and as a potential accelerator of evolution due to increased mutation rate. For Randseq1, 2, and 40 we also created strains from an ancestor in which the *mutS* was not deleted and after similar evolution scheme the exact same mutations arise within 2–5 weeks. In all strains, *lacI* was deleted (for all but the CRISPR/Cas9 strains, by spectinomycin resistance gene) and replaced by an extra double terminator (BioBricks BBa_B0015) to prevent transcription read through from upstream genes.

**Random sequences**. Each of the random sequences is 103 bp long, which is the same length as the WT lac intergenic region that was replaced. Also it is a typical length for an intergenic region in *E. coli* (the median intergenic region in *E. coli* is 134 bases long[33]). These random sequences (generated in Matlab) were used as starting sequences for promoter evolution because they represent the non-functional sequence space, without biases, as they contain no information. For the random sequences to also comply with GC content of the *E. coli* genome (50.8%) sequences with deviating GC content (lower than 45.6% or higher than 56.0%) were excluded. In addition, sequences with homo-nucleotide stretches longer than five were also excluded to avoid sequencing issues.

**Selection for lactose utilization**. Lab evolution was performed on liquid cultures grown on M9+GlyLac by daily dilution of 1:100 into 3 ml of fresh medium. Without the ability to utilize lactose the strains that carry random sequences reached a concentration of $3.3 \times 10^8$ cells/ml at saturation. Strains that had or acquired the ability to utilize lactose increased their final concentration to $\sim 1.5 \times 10^9$ cells/ml.

M9 base medium for 1 l included 100 μl $CaCl_2$ 1 M, 2 ml of $MgSO_4$ 1 M, 10 ml $NH_4Cl$ 2 M, 200 ml of M9 salts solution 5× (Sigma Aldrich). Concentrations of carbon source were 0.05% for glycerol and 0.2% for lactose for M9+GlyLac, 0.2% lactose for M9+Lac and 0.4% glycerol for M9+Gly (all in w/v).

Cultures were routinely checked for increased yield at saturation and samples were plated on M9+Lac plates for isolation of colonies that can utilize lactose as a sole carbon source. In parallel to our liquid M9+GlyLac selection for lactose-utilization we also performed agar-plate selection by growing random-sequence strains on non-selective medium (M9+Gly) and then plated them while in late logarithmic phase on M9+Lac plates to select for lactose-utilizing colonies.

All populations were evolved in parallel duplicates, but RandSeq1, 2, and 3, which evolved in four replicates.

**Quantifying growth and expression of the lac genes**. Growth curves were obtained by 24 h measurements of OD600 every 10 min. Expression of the lac genes was quantified by YFP florescence measurements. Both measurements performed by a Tecan M200 plate reader in 96-well plates. The expression of evolved cells was quantified by comparison to the control strain WTpromoter. All strains were measured for expression of the lac genes by YFP florescence by growth on M9 +Gly, except WTpromoter that was grown on M9+Lac for induction of the WT lac operon.

**Computational assessment of promoters in random sequences**. A sliding window was used to scan random sequences for promoters by counting the number of matches to the canonical promoter motifs. For each sequence window a promoter score was calculated according to a specific position weight matrix that contains a weight for each base in the −10 and −35 elements[35], including a weight for the length of the spacer.

***E. coli* genomic data**. Lists of essential genes and prophage genes were downloaded from EcoGene[33], a list of toxin–antitoxin gene couples was obtained from Eco-cyc[52], coding sequences of genes were downloaded from GeneBank (K-12 substr. MG1655, U00096).

**Recoding the coding sequence of *E. coli* genes**. To create alternative versions of the coding region, we recoded all translated genes in *E. coli* (n = 4261) 1000 different times while preserving the amino-acid sequence and codon bias[63]. As another null model we also shuffled the codons of each gene in 1000 permutations. Although a shuffled version of a gene does not preserve the amino-acid sequence, it exactly preserves the GC content of each gene, and thus it controls for another aspect that may result in accidental expression.

**Scoring putative accidental promoters in the coding region**. The output from the BPROM software[45,46] was used to evaluate putative accidental promoters in WT *E. coli* and its recoded versions. The software returns positions for all identified promoters in the input sequence; for each identified promoter an LDF score is provided—this score reflects how far the identified promoter was from the threshold of the linear discriminant function. For each identified promoter the software also returns weighted scores for the −10 and the −35 elements (TGn element score is integrated into the −10 element score). Our metric for putative accidental expression was calculated by summing the scores of the different promoter elements and then multiplying by the LDF score. This metric provided a proxy for expression from each of the identified

promoters. Since it is possible that multiple different promoters reside within the same coding sequence (either overlapping or distinct from one another) the putative accidental expression score reflects the sum of all promoters' scores in the calculated region. BPROM was run for each gene's coding sequence separately; therefore all scores were normalized to the gene's length to control for genes with different sizes; the scores are therefore shown as predicted expression per kb. Unless specified otherwise, the putative accidental expression scores include both promoters that occur in "sense" and "anti-sense" orientation, although for some analyses, like in the toxin–antitoxin analysis, the scores are spilt to "sense" and "anti-sense". The promoter scoring method mentioned here was validated by our experimental evolving library: the average score obtained for the 10% already active sequences was 460 ± 229 compared with an average of 85 ± 113 for the rest random sequences that were not active before evolution. Similarly, the average score of the evolved sequences went up from 85 ± 113 before evolution to 230 ± 149 after including the evolved mutations.

**Six-mer analysis**. Looking for depleted and overrepresented motifs, we counted the occurrences of all six-mers within the coding region of *E. coli*. We compiled a list of all 4096 possible six-mers and counted how many times each six-mer occurs in all WT coding region compared with the 1000 recoded versions. Then, we focused on six-mers that are significantly rare/abundant in WT version compared with their counting in the recoded versions

**Code availability**. Relevant code is available on https://github.com/AvihuYona/DenovoPromotersNatureComm2018. Other relevant codes are available from the authors.

**Data availability**. Relevant data are available from the authors.

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

## Acknowledgements

We thank the National Institute of Health for supporting this work and the Human Frontier Science Program for supporting A.H.Y. Special thanks for Idan Frumkin, Rebecca Herbst, and members of the Gorelab and the Almlab for fruitful discussions. We thank the Xie lab for providing strains and Gene-Wei Li, Jean-Benoit Lalanne, and Tami Lieberman for their helpful comments on the manuscript.

## Author contributions

A.H.Y., E.J.A., and J.G. designed the research. A.H.Y. performed the experiments and computational analyses. A.H.Y. and J.G. wrote the manuscript.

## Additional information

**Competing interests:** The authors declare no competing interests.

