## [Peer Review File · Nature Communications]

Reviewers' comments:

Reviewer #1 (Remarks to the Author):

Review of Yona et al.

In this work Yona et al. investigate the "evolvability" of bacterial promoters. They generate 40 random sequences and tested if these sequences can evolve to drive constitutive expression of the lac operon. 4/40 random sequences were active at the onset, and 23/40 became active after acquiring one mutation. A complimentary computational analysis showed that E. coli coding sequences are depleted of sequences predicted to have promoter activity.

The experiment is simple and well designed. The selection in a mixture of glycerol and lactose balanced a big adaptive advantage with some background growth. Integrating the construct into the chromosome avoided potential copy-number artifacts that can arise from plasmids. By remaking the evolved mutations in the parent sequences, the authors ruled out trans mutations and validated that the effects seen are true cis mutations for 16 mutants. The replicate evolutions enriched the findings, especially the example where two replicates evolved the same two mutations in the opposite order. Overall there are no major technical problems with the experiments.

This paper makes two interesting observations: 1) that random sequences can easily evolve into a minimal promoter with a single mutation and 2) that coding sequences in E. coli are depleted of sequences predicted to have promoter activity. These observations are well supported by the data and will be of interest to the transcriptional regulation and evolution communities.

The authors make a claim that transcription in E. coli is a metastable system with high evolvability. This is an interesting idea, but not something that is provable and should be left to the discussion.

A second claim is that the fitness landscape for a minimal promoter has a single peak (Line 192). This claim is supported by the evidence that all the evolved promoters resemble -10 and -35 motifs. However, only 40 sequences were tested, so the fitness landscape has only been sparsely sampled and may be more complex.

The paper would be strengthened by discussion of a few more ideas:

1) The authors should further justify their choice of equal base representation (Line 267) for their random sequences, as opposed to random sequences that preserve the dinucleotide frequencies of intergenic regions of E. coli. Random sequences with equimolar composition of the four bases are probably not sequences an E. coli would ever actually encounter.

2) The authors should probably discuss their conclusions in light of evolving a minimal, stripped down promoter. When might this minimal constitutive activity be adequate? How might this short evolutionary distance be the same or different for regulated or inducible promoters?

3) It would be useful to validate the predictive power of the BPROM algorithm by running it on the 40 random sequences. Does the algorithm predict the 4/40 active sequences? Do the evolved mutations change the algorithm predictions?

4) It might be interesting, but not required for publication, to develop a null model for the expected fraction of random sequences that are active or one mutation away from being active.

Minor points:

- Line 95, please clarify the last sentence of the paragraph.
- Line 175, please reference the supplementary discussion in this paragraph.
- Line 223, please reference the idea presented in this sentence.
- The legends of Figure 3 and S3 have extra hyphens.
- Line 234, please clarify whether polymerase collision refers to head on collisions or head to tail collisions or both. Consider referencing Figure S5 here.
- Line 252, this last sentence of the results is hard to follow.
- In the legend of Figure S3, please explain what the dots and corresponding numbers represent.
- Line 287, it might be useful to cite: Itzkovitz, S., & Alon, U. (2007). The genetic code is nearly optimal for allowing additional information within protein-coding sequences. *Genome Research*, 17(4), 405–412. <http://doi.org/10.1101/gr.5987307>
- Line 300, it might be useful to cite: Estrada, J., Ruiz-Herrero, T., Scholes, C., Wunderlich, Z., & DePace, A. H. (2016). SiteOut: An Online Tool to Design Binding Site-Free DNA Sequences. *PLoS ONE*, 11(3), e0151740. <http://doi.org/10.1371/journal.pone.0151740>

Reviewer #2 (Remarks to the Author):

Critique of "Random Sequences Rapidly Evolve into de novo Promoters" by Yona et al., submitted to Nature Communications.

23 December 2017

Note: This evaluation is a revised version of a previous evaluation I made for a different journal regarding an earlier version of this manuscript.

Summary

The manuscript by Yona et al. addresses an interesting question: how easy is it for bacterial promoters to evolve de novo? To address this question, the authors tested random 103 bp sequences for promoter activity in *Escherichia coli*. They found that ~10% of these sequences were functional. They further performed laboratory evolution experiments and observed that functional promoters could typically evolve out of these random starting sequences via just one point mutation. I think this paper is an important contribution to the literature on gene regulatory evolution, especially Fig. 3a. The experiments are simple, cleanly carried out, and give largely unambiguous results. I am not aware of any other work having addressed this problem so directly. That being said, I think this manuscript could be improved.

Primary points

The biggest weakness of this manuscript, as I see it, is a lack of discussion of whether these findings are quantitatively in line with what one would expect based on what is known about promoter composition in *E. coli*. In particular, this manuscript doesn't provide any calculation of how quickly one should expect promoters to arise in the evolution experiments that were performed. It should be relatively straightforward to do simulations that would provide approximate answers to this question using well-established position weight matrix models for RNAP with different -10/-35 hexamer spacings (Lisser and Margalit, 1993, ref. 21).

Secondly, the authors do not do any experimental validation of the hypothesized mechanism by which the evolved promoters end up working. It is not obvious that the creation of a strong RNAP site should be the only strategy evolution uses to create functional promoters de novo. Alternatively, binding sites for activating TFs might evolve instead, as the resulting recruitment of RNAP might be sufficiently strong to enable transcripts from a very weak RNAP binding site. I agree that the mutations observed in this work do suggest the creation of canonical RNAP binding sites. Indeed, the authors should point out that this is a nontrivial result. Still, it would be easy to further validate this hypothesized mechanism by demonstrating loss of promoter function upon a mutation to the essential adenine nucleotide at position -11 within putative -10 hexamers.

Finally, the claim that the observed mutations are reproducible from replicate to replicate is very surprising to me. If there is room, it might be nice to see more data in the main text supporting this claim. Still, the author's claim that "The expression landscape for promoters in this environment therefore appears to be single-peaked" (lines 191-192) clearly isn't right if one thinks of the landscape as spanning all 103-mers since different starting sequences don't all evolve to the same final promoter sequence. I would like to see a more careful discussion of what these results do and do not show.

Secondary points

The authors might want to be cautious in their discussion of "accidental expression." No actual measurements of accidental expression were performed, only a statistical analysis of gene sequences. Much of what the authors call accidental expression could simply reflect inaccuracies in the BPROM model (ref 28) that was used to make predictions. The term "putative accidental expression" might be more appropriate.

Also, it is not clear why the authors rely on BPROM instead of just scanning the established RNAP PWMs across sequences. An analysis using the latter strategy would be a lot easier to interpret, and I'd be surprised if it gave substantively different results. At the very least, the authors should give a more complete description of precisely how they used BPROM. For instance, are scores for promoters of different lengths comparable? How and how does BPROM integrate scores across an entire 103bp region?

Fig. 4a is kind of misleading. I think supplemental Fig. 3b more accurately conveys the kind of information the authors mean to display here.

Reviewer #3 (Remarks to the Author):

The authors of the manuscript try to solve the question of how new promoters can arise de novo from random sequences.

The manuscript has two parts, which are thematically related, but each has their own particular shortcomings. The first is an experimental inspection of the evolvability of "random" sequences, while the second is a computational survey of the E.coli genome in an attempt to make a mechanistic claim about the impact of the spontaneous evolvability of promoters in the genome.

I must say, that the results are not very surprising given prior work, and that similar (although not identical) studies have been performed. In this respect, the authors are able to identify and confirm the following:

- A random sequence that has motifs close to a promoter, will likely mutate and gain promoter-like abilities.

- Lots of random sequences ALREADY have promoter motifs, and plenty more have motifs one or two mutations away.

In this respect, the manuscript provides insight and confirmation about the mechanistic nature of bacterial motif detection, but sheds little light into the matter of how a sequence that is *far* from a motif is able to gain promoter activity.

While this might sound trivial, I think it is important that the authors explicitly recognize that their random sequences are close to promoters, and that this is what makes them likely to gain transcription.

The second part of the manuscript is highly speculative, and its validity rests on the assumption that the accidental expression metric the authors use is appropriate. I don't see anywhere how this metric is defined (other than by a multiplication of score outputs and probability scores). Most of the discussion relies on this, as well as most of the narrative aspect of the manuscript (the term accidental is found all over).

While the proposed metric sounds interesting, I see no evidence one could peg to it. The depletion in motifs might mean many things, and could be confounding in what the authors state as accidental expression. What evidence do the authors have for this? Could there be selection against other features that the authors could be mistakenly be calling against accidental expression?

Can this metric be applied to the variety of E.coli genomes and transcriptomes, and inspected for its validity in regions where polymorphisms cause variation of "accidental expression" scores and actual transcription readout?

Can this be explained in other bacterial systems? This is important because the authors often explain their interpretations in evolutionary terms.

To summarize, I think the authors raise interesting points, but the strategies are somewhat incomplete. I think the first part of the manuscript is fine as it is and it provides interesting results, as long as the authors explicitly state that what they are measuring is how short mutational distance of any sequence to a promoter affects its promoter like effects.

The second part is a bit more problematic, and while interesting as a correlational study, I don't think it has enough support for the vast amount of interpretation the authors go into.

We thank the editor and referees for their careful reading of our manuscript. Below (in blue) we describe how we have modified the manuscript to address the comments raised (in black).

Reviewer #1 (Remarks to the Author):

Review of Yona et al.m

In this work Yona et al. investigate the “evolvability” of bacterial promoters. They generate 40 random sequences and tested if these sequences can evolve to drive constitutive expression of the lac operon. 4/40 random sequences were active at the onset, and 23/40 became active after acquiring one mutation. A complimentary computational analysis showed that E. coli coding sequences are depleted of sequences predicted to have promoter activity.

The experiment is simple and well designed. The selection in a mixture of glycerol and lactose balanced a big adaptive advantage with some background growth. Integrating the construct into the chromosome avoided potential copy-number artifacts that can arise from plasmids. By remaking the evolved mutations in the parent sequences, the authors ruled out trans mutations and validated that the effects seen are true cis mutations for 16 mutants. The replicate evolutions enriched the findings, especially the example where two replicates evolved the same two mutations in the opposite order. Overall there are no major technical problems with the experiments.

This paper makes two interesting observations: 1) that random sequences can easily evolve into a minimal promoter with a single mutation and 2) that coding sequences in E. coli are depleted of sequences predicted to have promoter activity. These observations are well supported by the data and will be of interest to the transcriptional regulation and evolution communities.

>>We thank the referee for the nice summary of our results and the positive assessment of their novelty.

The authors make a claim that transcription in E. coli is a metastable system with high evolvability. This is an interesting idea, but not something that is provable and should be left to the discussion.

>> We agree. This notion was omitted.

A second claim is that the fitness landscape for a minimal promoter has a single peak (Line 192). This claim is supported by the evidence that all the evolved promoters resemble -10 and -35 motifs. However, only 40 sequences were tested, so the fitness landscape has only been sparsely sampled and may be more complex.

>> This is a valid point. This claim about the fitness landscape was taken out.

The paper would be strengthened by discussion of a few more ideas:

1) The authors should further justify their choice of equal base representation (Line 267) for their random sequences, as opposed to random sequences that preserve the dinucleotide frequencies of intergenic regions of E. coli. Random sequences with equimolar composition of the four bases are probably not sequences an E. coli would ever actually encounter.

>> We agree with the reviewer that our choice to generate random sequences composed of equal proportions of all four bases should be better justified. We added text that clarifies our choice (lines 44-48).

Our choice to generate random sequences composed of equal proportions of all four bases was in order to obtain starting sequences for evolution that contain no information. Since the dinucleotide composition of the E.coli genome is close to 50% (50.8%), these random sequences also comply with E.coli’s GC content, meaning that in terms of dinucleotide composition they represent a general case of a silent gene located somewhere in the genome.

Starting sequences that are made according to the dinucleotide composition of E.coli’s intergenic region have significantly higher AT content 56.7% AT compared to the rest of the genome 48.2% AT. Since the promoter motifs in E.coli are AT-rich – the two main consensus elements are 10/12 AT, and the additional up elements are either 6/9 or 11/11 AT; when inactive genes occur downstream to intergenic regions they probably have higher chances to become active. These differences in AT-content make it more probable for

silent genes that reside in intergenic regions to get activated. Therefore, testing random sequences that were generated according to dinucleotide frequencies of the intergenic region is likely to yield even more active promoters than random sequences generated according to the overall dinucleotide frequencies of the entire E.coli genome.

In addition, our aim to study de novo activation of inactive genes was designed according to the scenarios that typically lead to silent genes in the genome - namely due to HGT events or due to genomic rearrangements that displace intergenic regions that harbor native promoters. For both cases we cannot assume that the location of the inactive genes is biased to be adjacent to an intergenic region as intergenic regions comprise only ~12% of the E.coli genome. Therefore, we composed random sequences according to the overall dinucleotide frequencies in E.coli (50.8% GC) merely as a way to capture the general case of de novo gene activation.

Here is the modified text:

“Using purely random sequences as a starting point for promoter evolution is especially suitable for genomes with ~50% GC content, like the E. coli genome which is 50.8% GC. For such genomes, random sequences can serve as a null model when testing for functionality without introducing biases or confounding factors due to deviating from the natural GC content of the studied genome.”

2) The authors should probably discuss their conclusions in light of evolving a minimal, stripped down promoter. When might this minimal constitutive activity be adequate? How might this short evolutionary distance be the same or different for regulated or inducible promoters?

>> We indeed thought about the evolutionary role of such basic constitutive promoters and we agree that discussing this in the text will benefit the readers, therefore we added text that refer to this point (lines 313-319).

The promoters evolved in our experiments generated ~50% of the expression produced from a fully induced lac promoter with only one mutation and promoters that further evolved another mutation even exceeded the lac promoter expression. Yet, these evolved promoters have no regulation and their expression is constitutive until the cells get to stationary phase. Resource depletion and sigma factor changes (from the primary sigma-70 to the stationary phase sigma-38) are likely the only factors that inhibit their activity.

We hypothesize that promoters such as the ones evolve in our library play a role in the evolution of the transcriptional network. First, new genes that confer antibiotic resistance or a novel ability to exploit nutrients from the environment, not necessarily require regulated/induced expression, and indeed many of known antibiotic resistance genes are constitutively expressed. Second, such unregulated promoters can serve as an evolutionary stepping-stone until additional mutations refine their expression pattern to further increase the fitness.

The length of a regulatory binding sites is typically 6-10bp, not very different from the basic promoter elements that together comprise 12bp. Therefore, we predict that for many transcription regulators, such a stepwise sequence of mutations exists so that binding site can evolve. Similar to the promoter evolution, here too, it is likely that a single mutation can only confer partial functionality while subsequent mutations might be needed to fully capture the binding site. However, while in our promoter scenario the sequence search space was ~100bp, in the case of evolving a subsequent binding site the sequence space might be shorter. In order for the regulator to have an effect it usually has to be located either very close to the promoter sites, or in between the promoter and the gene's coding sequence, for example in the case of repressors.

Here is the modified text:

“Despite generating expression levels similar to the WT lac promoter, the promoters evolved in our library are of very low complexity, as most of the activating mutations involved no additional factors but the two basic promoter motifs. Although the evolved promoters likely have no regulation, we hypothesize that such crude promoters might play an important role in the evolution of the transcriptional network, as newly activated genes do not necessarily require regulated/induced expression in order to confer a significant

advantage. Furthermore, such stripped down promoters can serve as an evolutionary stepping-stone until regulation evolves, perhaps also by stepwise point mutations.”

3) It would be useful to validate the predictive power of the BPROM algorithm by running it on the 40 random sequences. Does the algorithm predict the 4/40 active sequences? Do the evolved mutations change the algorithm predictions?

>> Indeed, the BPROM algorithm yields an average score of 460 ± 229 for the 4/40 active random sequences compared with an average of 85 ± 113 for the rest random sequences that weren't active before evolution. After evolution the scores average of the mutated sequences went up from 85 ± 113 to 230 ± 149 . This data was now added to text (lines 428-432)

Here is the modified text:

“The promoter scoring method mentioned here was validated by our experimental evolving library: the average score obtained for the 10% already active sequences was 460 ± 229 compared with an average of 85 ± 113 for the rest random sequences that were not active before evolution. Similarly, the average score of the evolved sequences went up from 85 ± 113 before evolution to 230 ± 149 after including the evolved mutations.”

4) It might be interesting, but not required for publication, to develop a null model for the expected fraction of random sequences that are active or one mutation away from being active.

>> We thank the reviewer for this comment. We now have an additional computational analysis that addresses this comment as well as a new paragraph and a new figure to describe and discuss the new results. In order to obtain an estimate on the fractions of random sequences that are active or one mutation away from being active we generated 30,000 new random sequences (the same way the 40 experimental random sequences were generated) and for each sequence we assessed its ability to function as a promoter, before and after in-silico evolution. We use two different benchmarks to predict if a sequence (before or after evolution) contains an active promoter:

- 1) If the find in the sequence a promoter structure that captures the six most significant bases of the canonical promoter's consensus sequences - the TTGnnn from the TTGACA motif, and the TAnnnT from the TATAAT (with a valid spacer).
- 2) If the random sequence get a score as the median E.coli constitutive promoter (or higher) when the score is calculated according to the known position specific matrix of the canonical promoter (without using BPROM).

For both benchmarks we obtained results that are in agreement with our experimental results (see new Figure 4 and lines 205-218).

Here is the modified text:

“Computational assessment of promoter sequence accessibility

Our evolution experiment showed that a single mutation could often produce expression levels similar in magnitude to the expression level produced by the WT lac promoter. To get a numerical perspective on these findings, we assessed the mutational distance that separates random sequences from the canonical promoter of E. coli. To this end, we computationally created 30,000 random sequences (the same way the experimental RandSeq1 to 40 were generated) and ran a template of the canonical promoter against their sequences, using a sliding window. Since the importance of each base for promoter activity differs considerably, we weighted each base according to the position-specific matrix of the E. coli canonical promoter (Methods). In the same way we also obtained scores for the 556 constitutive promoters³⁹ of E. coli and set their median score as the benchmark that qualifies as a promoter (Supplementary Note 3). The results from this analysis showed that the fraction of random sequences that qualify as promoters and the fraction that are one mutation away from a promoter coincide with the fractions observed in our experimental library. Similar fractions were also observed when the benchmark for a promoter was set to capture the core bases of the canonical motifs (TTGnnn and TAnnnT, where ‘n’ represent any base)(Fig. 4).”

Minor points:

- Line 95, please clarify the last sentence of the paragraph.
>> Sentence was rewritten.
- Line 175, please reference the supplementary discussion in this paragraph.
>> Reference was added.
- Line 223, please reference the idea presented in this sentence.
>> Reference was added.
- The legends of Figure 3 and S3 have extra hyphens.
>> Corrected.
- Line 234, please clarify whether polymerase collision refers to head on collisions or head to tail collisions or both. Consider referencing Figure S5 here.
>> “head on and head-to-tail” collisions specified.
- Line 252, this last sentence of the results is hard to follow.
>> Sentence was rewritten.
- In the legend of Figure S3, please explain what the dots and corresponding numbers represent.
>> Thank you for this remark; the figure legend now includes all details, including the indication that the dots and the numbers of each plot represent the mean value.
- Line 287, it might be useful to cite: Itzkovitz, S., & Alon, U. (2007). The genetic code is nearly optimal for allowing additional information within protein-coding sequences. *Genome Research*, 17(4), 405–412.
<http://doi.org/10.1101/gr.5987307>
>> Citation was added.
- Line 300, it might be useful to cite: Estrada, J., Ruiz-Herrero, T., Scholes, C., Wunderlich, Z., & DePace, A. H. (2016). SiteOut: An Online Tool to Design Binding Site-Free DNA Sequences. *PLoS ONE*, 11(3), e0151740.
<http://doi.org/10.1371/journal.pone.0151740>
>> Citation was added.

Reviewer #2 (Remarks to the Author):

Critique of “Random Sequences Rapidly Evolve into de novo Promoters” by Yona et al., submitted to Nature Communications.

23 December 2017

Note: This evaluation is a revised version of a previous evaluation I made for a different journal regarding an earlier version of this manuscript.

Summary

The manuscript by Yona et al. addresses an interesting question: how easy is it for bacterial promoters to evolve de novo? To address this question, the authors tested random 103 bp sequences for promoter activity in *Escherichia coli*. They found that ~10% of these sequences were functional. They further performed laboratory evolution experiments and observed that functional promoters could typically evolve out of these random starting sequences via just one point mutation. I think this paper is an important contribution to the literature on gene regulatory evolution, especially Fig. 3a. The experiments are simple, cleanly carried out, and give largely unambiguous results. I am not aware of any other work having addressed this problem so directly. That being said, I think this manuscript could be improved.

Primary points

The biggest weakness of this manuscript, as I see it, is a lack of discussion of whether these findings are quantitatively in line with what one would expect based on what is known about promoter composition in *E. coli*. In particular, this manuscript doesn't provide any calculation of how quickly one should expect promoters to arise in the evolution experiments that were performed. It should be relatively straightforward to do simulations that would provide approximate answers to this question using well-established position weight matrix models for RNAP with different -10/-35 hexamer spacings (Lisser and Margalit, 1993, ref. 21).

>> We thank the reviewer for this comment. We performed new analyses and added a full paragraph and a new figure in which we discuss the new results (see new Figure 4 and lines 205-218).

We used a position weight matrix (according to the reference suggested by the reviewer) to calculate a promoter score for 30,000 new random sequences (generated in a similar way to the 40 experimental random sequences). We also obtained scores for the known 556 constitutive promoters in *E. coli* and compared the scores distribution of the native promoters to those of the random sequences. Next, we performed in-silico evolution for each of the random sequences separately, looking for the best single mutation in terms of increasing the promoter score. The new distribution, of the in-silico evolved random sequences, shows results that coincide with our experimental results – a random sequence of ~100 bases typically needs a single mutation in order to create a promoter with a score that is similar to scores of native constitutive promoter.

Here is the modified text:

“Computational assessment of promoter sequence accessibility

Our evolution experiment showed that a single mutation could often produce expression levels similar in magnitude to the expression level produced by the WT lac promoter. To get a numerical perspective on these findings, we assessed the mutational distance that separates random sequences from the canonical promoter of *E. coli*. To this end, we computationally created 30,000 random sequences (the same way the experimental RandSeq1 to 40 were generated) and ran a template of the canonical promoter against their sequences, using a sliding window. Since the importance of each base for promoter activity differs considerably, we weighted each base according to the position-specific matrix of the *E. coli* canonical promoter (Methods). In the same way we also obtained scores for the 556 constitutive promoters³⁹ of *E. coli* and set their median score as the benchmark that qualifies as a promoter (Supplementary Note 3). The results from this analysis showed that the fraction of random sequences that qualify as promoters and the fraction that are one mutation away from a promoter coincide with the fractions observed in our experimental library. Similar fractions were also observed when the benchmark for a promoter was set to capture the core bases of the canonical motifs (TTGnnn and TAnnnT, where 'n' represent any base)(Fig. 4).”

Secondly, the authors do not do any experimental validation of the hypothesized mechanism by which the evolved promoters end up working. It is not obvious that the creation of a strong RNAP site should be the only strategy evolution uses to create functional promoters de novo. Alternatively, binding sites for activating TFs might evolve instead, as the resulting recruitment of RNAP might be sufficiently strong to enable transcripts from a very weak RNAP binding site. I agree that the mutations observed in this work do suggest the creation of canonical RNAP binding sites. Indeed, the authors should point out that this is a nontrivial result. Still, it would be easy to further validate this hypothesized mechanism by demonstrating loss of promoter function upon a mutation to the essential adenine nucleotide at position -11 within putative -10 hexamers.

>> We agree with the reviewer comment. Indeed it is nontrivial that the vast majority of promoters evolved via creating/strengthening RNAP binding sites. Alternatively a mutation that create/strengthen a binding site of a transcription factor might also result in de-novo expression, when a preexisting weak promoter motif exist nearby (see for example Randseq 29 and 40). This is especially relevant since we see that the chances of having weak promoter motifs in random sequences are not that low. We also note that running BPROM, which also predict transcription-binding sites nearby promoters, on the evolved sequences did not retrieve indications for mutations that created/strengthened binding sites of transcription factors. We now address this comment in the text (lines 302-311).

Here is the modified text:

“The activating mutations found in our experiments suggest that new promoters largely emerge by creating the canonical RNAP binding sites (TATAAT and TTGACA). It is not obvious why the alternative strategy of evolving binding sites for transcription factors has not been widely observed in our library. However, in two cases, RandSeq29 and RandSeq40, we observed that predicted promoters in the random sequences were not active before evolution until activated mutations occurred downstream to their TATAAT element, but did not create a promoter motif (see Supplementary Table 1). In these two promoters, it is possible that the evolved mutations (which occurred repeatedly) either strengthen a transcription-activator binding site or reduced the affinity of a repressor that blocked expression from the predicted promoter upstream. Especially since in both cases two different mutations evolved next to one another, which might indicate targeting a single transcription factor binding-site.”

In addition, we artificially mutated the evolved promoters presented in Figure 2 by changing the essential adenine nucleotide in position -11 to guanine. Indeed, mutating the -11 adenine completely abolished the expression of RandSeq 1 and 3. In the case of RandSeq2 in which the evolved promoter was overlapping with another predicted promoter, we only mutated the -11 adenine of the evolved promoter (and not of the additional overlapping one). In this mutant, low expression was still detected yet expression dropped by 3.2 fold compared to the evolved RandSeq2 (see 138-146).

Here is the modified text:

“To validate that the evolved mutations indeed induced expression by creating a canonical promoter we demonstrated loss of promoter function upon mutating the most essential position within the evolved canonical promoter - the adenine in position -11. Changing the -11 adenine to guanine completely abolished the expression of RandSeq1 and RandSeq3. Yet, in the case of RandSeq2 the evolved promoter overlaps with another predicted promoter (see Supplementary Table 1); therefore we only mutated the -11 adenine of the evolved promoter. In this mutant, low expression was still detected yet expression dropped by 3.2-fold compared to the evolved RandSeq2. The loss of expression in these three mutated sequences is therefore consistent with our interpretation that the evolved solutions had indeed created the canonical promoter motifs.”

Finally, the claim that the observed mutations are reproducible from replicate to replicate is very surprising to me. If there is room, it might be nice to see more data in the main text supporting this claim. Still, the author’s claim that “The expression landscape for promoters in this environment therefore appears to be single-peaked” (lines 191-192) clearly isn’t right if one thinks of the landscape as spanning all 103-mers since different starting sequences don’t all evolve to the same final promoter sequence. I would like to see a more careful discussion of what these results do and do not show.

>> We thank the reviewer of this comment. Our claim that the observed mutations are reproducible was intended only with regard to the first random sequences (RandSeq1,2, and 3) in Figure 2, and only in an informative manner for making the figure more comprehensible.

Indeed for the whole library it is not quite the case. For ~1/3 of the strains the different replicates showed different activating mutations, and ~1/2 of the strains that did show the same mutation in the two replicates were deletions/insertions that bring promoter motifs together (summarized in the mutations table). For deletions/insertions, different mutations are more likely to occur only after the promoter parts are put together. In addition, we understand the problem in the claim about the single-peaked landscape and we therefore omitted it from the manuscript.

We add a short discussion about some parallel evolution we observed within the different replicates of some random sequences (lines 197-203).

Here is the modified text:

“A significant fraction of the strains in our evolved library (~2/3) showed parallel evolution i.e. the same activating mutations occurred in the different population replicates. This indicates that a ~100-base random sequence typically does not include multiple segments that can evolve into a promoter by a single mutation. We also saw parallel evolution in the random sequences that evolved by multiple stepwise mutations, yet

the mutations sometimes occurred in a different order (like in RandSeq2, see Supplementary Table 1). Interestingly, these stepwise mutations show no signs of epistasis, as their contribution to the expression level is largely additive.”

Secondary points

The authors might want to be cautious in their discussion of “accidental expression.” No actual measurements of accidental expression were performed, only a statistical analysis of gene sequences. Much of what the authors call accidental expression could simply reflect inaccuracies in the BPROM model (ref 28) that was used to make predictions. The term “putative accidental expression” might be more appropriate.

>> We agree that using this terminology might confuse the readers and that a better term is “putative accidental expression”. We now specifically state (lines 233-238) that our analyses on accidental expression are merely predictions, based on sequence analysis, and that we interpret such predictions as “putative accidental expression”

Here is the modified text:

“To test if the *E. coli* genome avoids accidental promoters from occurring inside genes, we computationally scanned the WT genome and identified putative promoters within the coding region. In order to assess whether the WT genome of *E. coli* has minimized accidental promoters we also scanned a thousand alternative versions of the *E. coli* genome (generated in-silico) and compared their accidental promoters to those of the WT genome. We use the term “accidental expression” to describe putative expression from promoters that were predicted by sequence analysis of the coding region.”

Also, it is not clear why the authors rely on BPROM instead of just scanning the established RNAP PWMs across sequences. An analysis using the latter strategy would be a lot easier to interpret, and I’d be surprised if it gave substantively different results. At the very least, the authors should give a more complete description of precisely how they used BPROM. For instance, are scores for promoters of different lengths comparable? How and how does BPROM integrate scores across an entire 103bp region?

>> We do realize that our description on the use of BPROM should be more detailed and we now addressed this point as well adding a description of an alternative analysis to the BPROM in which we scan the WT genome for all possible motifs and compare the results to the recoded versions. Reassuringly, this alternative analysis results in conclusions similar to those of the BPROM (see lines 246-260 and 415-433).

In our experimental library none of the activating observed mutations created/improved other promoter motifs rather than the -10 and the -35, like the UP element or the TGn motif. Nonetheless, when we were to estimate putative accidental expression from the *E. coli* genome we aimed for a software that takes all known motifs into account and not only the two major ones. We agree that scanning the genome with PWMs of the two major motifs, the -10 and the -35, will largely give similar results to such software. Nonetheless a programed scan of the genome that factors in all features that affect expression seemed like the optimal approach, simply because it takes into account more genetic features that may affect transcription. Furthermore, since such software dedicated for this purpose are already available we thought it would be better to use them rather than to develop one in-house. We chose BPROM merely because it was the only software, which the developers allowed us batch runs from a script (for other software we could only run limited amount of queries and on an a user-required website and not via scripts).

In terms of our use of BPROM, we now specify how did we use BPROM to assess putative accidental expression across the *E.coli* genome:

1. Since it is possible that multiple different promoters reside within the same coding sequence (either overlapping or distinct from one another) the predicted expression is the sum of all promoters scores in the calculated region.
2. We run BPROM for each gene’s coding sequence separately. To control for genes with different sizes, all scores are normalized to the gene’s length and the scores are therefore shown as predicted expression per Kb.

3. Unless specified otherwise the score of putative accidental expression includes both promoters that occur in 'sense' and 'anti-sense' orientation, although for some analyses we split the score to 'sense' and 'anti-sense' like in the toxin-antitoxin analysis.

Few more details on the way we use BPROM are written below in the response to reviewer #3

Here is the modified text:

“Our main method to identify promoters inside genes was based on BPROM^{45,46}, an available software for promoter detection in *E. coli* that takes into account not only the -10 and -35 elements but also other factors that affect transcription level, like the TGn element and transcription factor binding sites, as well as the spacing between the elements (Methods and Supplementary Note 4). In addition, we performed a “six-mer” analysis - a straightforward unbiased analysis in which we count the number of occurrences for all possible six-mer motifs across the genome. Comparing the differences in six-mer occurrences between the WT genome to the recoded ones revealed which motifs are overrepresented and which are underrepresented in the WT *E. coli* genome (Methods).

We used both of these methods to compare predicted accidental promoters of the WT *E. coli* with its thousand recoded versions and found that both methods predict significantly less accidental expression in the WT *E. coli*. In other words, among all the possible genomic versions that can encode the *E. coli* proteome, the WT version seems to be one of the lowest in terms of accidental expression (Fig. 5a, Supplementary Fig. 3 and Supplementary Table 2). The *E. coli* genome has therefore likely been under selection to decrease accidental expression within the coding region of genes.”

“**Scoring putative accidental promoters in the coding region** – the output from the BPROM software^{45,46} was used to evaluate putative accidental promoters in WT *E. coli* and its recoded versions. The software returns positions for all identified promoters in the input sequence; for each identified promoter an LDF score is provided - this score reflects how far the identified promoter was from the threshold of the linear discriminant function. For each identified promoter the software also returns weighted scores for the -10 and the -35 elements (TGn element score is integrated into the -10 element score). Our metric for putative accidental expression was calculated by summing the scores of the different promoter elements and then multiplying by the LDF score. This metric provided a proxy for expression from each of the identified promoters. Since it is possible that multiple different promoters reside within the same coding sequence (either overlapping or distinct from one another) the putative accidental expression score reflects the sum of all promoters' scores in the calculated region. BPROM was run for each gene's coding sequence separately; therefore all scores were normalized to the gene's length to control for genes with different sizes; the scores are therefore shown as predicted expression per Kb. Unless specified otherwise the putative accidental expression scores include both promoters that occur in 'sense' and 'anti-sense' orientation, although for some analyses, like in the toxin-antitoxin analysis, the scores are split to 'sense' and 'anti-sense'. The promoter scoring method mentioned here was validated by our experimental evolving library: the average score obtained for the 10% already active sequences was 460 ± 229 compared with an average of 85 ± 113 for the rest random sequences that were not active before evolution. Similarly, the average score of the evolved sequences went up from 85 ± 113 before evolution to 230 ± 149 after including the evolved mutations.”

Fig. 4a is kind of misleading. I think supplemental Fig. 3b more accurately conveys the kind of information the authors mean to display here.

>> We agree to this point and Fig. 4a was changed accordingly to present the data in a similar way to supplementary Fig. 3b

Reviewer #3 (Remarks to the Author):

The authors of the manuscript try to solve the question of how new promoters can arise de novo from random sequences.

The manuscript has two parts, which are thematically related, but each has their own particular

shortcomings. The first is an experimental inspection of the evolvability of "random" sequences, while the second is a computational survey of the E.coli genome in an attempt to make a mechanistic claim about the impact of the spontaneous evolvability of promoters in the genome.

I must say, that the results are not very surprising given prior work, and that similar (although not identical) studies have been performed. In this respect, the authors are able to identify and confirm the following:

- A random sequence that has motifs close to a promoter, will likely mutate and gain promoter-like abilities.

- Lots of random sequences ALREADY have promoter motifs, and plenty more have motifs one or two mutations away.

In this respect, the manuscript provides insight and confirmation about the mechanistic nature of bacterial motif detection, but sheds little light into the matter of how a sequence that is *far* from a motif is able to gain promoter activity. While this might sound trivial, I think it is important that the authors explicitly recognize that their random sequences are close to promoters, and that this is what makes them likely to gain transcription.

>> We agree to this comment and in fact we now better emphasize this as one of the points of our paper. We now discuss this point while emphasizing the following:

- 1) Random sequences are not so far away from acting as promoters (i.e. 60% of ~100bp random sequence is one mutation away, and 10% are already active)
- 2) When a sequence (not necessarily a random sequence) requires two or more mutations to evolve an active promoter it is not likely that the cells would evolve expression by a series of mutations. In such cases it is more likely that other solution would be used, like copying an existing promoter by genomic rearrangements. This is because the chance of acquiring two mutations (when the first mutation is not beneficial) is much lower and thus requires much longer time.

In our library, ~30% of random sequences could not evolve by mutations in the random sequences, yet they could evolve in other ways, like by copying existing promoters. For example, we show for RandSeq27, which none of its replicates evolved by mutations in the random sequence, that there are two point mutations that when act together create an active promoter. To this end, we created a modified version of RandSeq27 that includes an artificial mutation obtained by in-silico evolution of RandSeq27. When this mutation was introduced artificially to RandSeq27 it was not sufficient to induce promoter activity. Yet, after evolving this modified RandSeq27, a second mutation created a promoter together with the artificial mutation.

- 3) The coding sequences of the wild-type E.coli genes show lower probabilities to include active promoters when compared with other alternative versions of themselves achieved either by recoding their amino-acid sequences or by shuffling their codons. We attribute this result to the interference caused when promoters within the coding regions hamper the expression from the native promoters.

Here are the texts that address the points above:

Here is the modified text (lines 341-345):

“Tuning the promoter recognition machinery to such a low specificity so that one mutation is often sufficient to induce substantial expression is crucial for the ability to evolve de novo promoters. If two or more mutations were needed in order to create a promoter, cells would face a much greater fitness-landscape barrier that would drastically reduce their ability to evolve promoters de novo. In such cases cells are likely to copy existing promoters via genomic rearrangements.”

Lines 297-299:

“Our study suggests that the sequence recognition of the transcription machinery is rather permissive and not restrictive to the extent that the majority of non-specific sequences are on the verge of operating as active promoters.”

See also Supplementary Note 2 (not attached because too long).

The second part of the manuscript is highly speculative, and its validity rests on the assumption that the accidental expression metric the authors use is appropriate. I don't see anywhere how this metric is defined (other than by a multiplication of score outputs and probability scores). Most of the discussion relies on this, as well as most of the narrative aspect of the manuscript (the term accidental is found all over).

>> We thank the reviewer for this comment. We do realize that our description of the predicted score for accidental expression was not detailed enough, and that our use of BPROM should be explained in more details. We now addressed this comment (see lines 246-260 and 415-433).

Our experimental library showed that active promoters frequently occur in random sequences i.e. that a typical random sequence of 1kb should already include an active promoter that can drive expression equivalent to ~50% of a fully induced WT lac operon. Therefore, we asked how many times promoters occur in the genome in places where they should not normally occur – i.e. not upstream to a gene, but inside the coding sequence of a gene. Such promoters are now defined in the manuscript as a proxy for putative accidental expression.

We use the term accidental expression merely to reflect that when such promoters occur inside a gene they may interfere with its expression due to RNAP interference caused by interactions between the “normal” elongating RNAP coming from the promoter outside the gene, to the RNAP that comes from within the gene (this notion has been recently shown experimentally by Brophy & Voigt 2016). Such interference can occur due to collisions (head-to-head or head-to-tail), but it can also be the result of the accidental RNAP attached in the middle of the gene without elongating (a.k.a sitting duck) which interferes with the elongating RNAP that comes from the original gene promoter.

In order for such a metric to be meaningful one must compare the results obtained on the WT genome to alternative versions that go through the same calculation. These versions (a thousand of them) were obtained according to two null models, each controlling for a different confounding factors (described in our response the next comment below).

To assess putative active promoters from sequences we needed an algorithm that takes into account all known promoter motifs (not only the two major ones that were highlighted by our experimental results). Since such softwares dedicated for that purpose already exist, we used BPROM – a software used in more than 800 publications, which allows promoter detection in batch runs from scripts (unlike other softwares for which one can only run limited amount of queries and while using a user-required website and not via scripts).

In terms of our use of BPROM, we now provide additional details on how we used BPROM to assess putative accidental expression across the E.coli genome:

1. The BPROM software gets an input sequence and returns all identified promoters and their positions within the input sequence. For each identified promoter an LDF score is provided - this score reflect how far the identified promoter was from the threshold of the linear discriminant function. The higher the value the higher the confidence that the identified promoter is indeed an active promoter. The BPROM also returns for each promoter identified a weighted score for the -10 element, and for the -35 element are given. These scores reflect how close the elements are to the conserved promoter elements of E. coli, including additional promoter elements like the TGn element.
2. The putative accidental expression metric that we assign for identified promoter is the sum of the scores given for the elements multiplied by the LDF score. In this way we have a proxy for expression for each identified promoter. For example a promoter with good elements score, but with low LDF score, for example due to a non optimal spacer, will get an overall low expression score, compared to a promoter with a similar elements score but also with an optimal spacer.

3. Since it is possible that multiple different promoters reside within the same coding sequence (either overlapping or distinct from one another) the predicted expression is the sum of all promoters scores in the calculated region
4. We run BPROM for each gene's coding sequence separately. To control for genes with different sizes, all scores are normalized to the gene's length and the scores are therefore shown as predicted expression per Kb.
5. Unless specified otherwise the score of putative accidental expression includes both promoters that occur in 'sense' and 'anti-sense' orientation, although for some analyses the scores are split to 'sense' and 'anti-sense' like in the toxin-antitoxin analysis.

Here is the modified text:

“Our main method to identify promoters inside genes was based on BPROM, an available software for promoter detection in E. coli that takes into account not only the -10 and -35 elements but also other factors that affect transcription level, like the TGn element and transaction factor binding sites, as well as the spacing between the elements (Methods and Supplementary Note 4). In addition, we performed a “six-mer” analysis - a straightforward unbiased analysis in which we count the number of occurrences for all possible six-mer motifs across the genome. Comparing the differences in six-mer occurrences between the WT genome to the recoded ones revealed which motifs are overrepresented and which are underrepresented in the WT E. coli genome (Methods).

We used both of these methods to compare predicted accidental promoters of the WT E. coli with its thousand recoded versions and found that both methods predict significantly less accidental expression in the WT E. coli. In other words, among all the possible genomic versions that can encode the E. coli proteome, the WT version seems to be one of the lowest in terms of accidental expression (Fig. 5a, Supplementary Fig. 3 and Supplementary Table 2). The E. coli genome has therefore likely been under selection to decrease accidental expression within the coding region of genes.”

“**Scoring putative accidental promoters in the coding region** – the output from the BPROM software was used to evaluate putative accidental promoters in WT E. coli and its recoded versions. The software returns positions for all identified promoters in the input sequence; for each identified promoter an LDF score is provided - this score reflects how far the identified promoter was from the threshold of the linear discriminant function. For each identified promoter the software also returns weighted scores for the -10 and the -35 elements (TGn element score is integrated into the -10 element score). Our metric for putative accidental expression was calculated by summing the scores of the different promoter elements and then multiplying by the LDF score. This metric provided a proxy for expression from each of the identified promoters. Since it is possible that multiple different promoters reside within the same coding sequence (either overlapping or distinct from one another) the putative accidental expression score reflects the sum of all promoters' scores in the calculated region. BPROM was run for each gene's coding sequence separately; therefore all scores were normalized to the gene's length to control for genes with different sizes; the scores are therefore shown as predicted expression per Kb. Unless specified otherwise the putative accidental expression scores include both promoters that occur in 'sense' and 'anti-sense' orientation, although for some analyses, like in the toxin-antitoxin analysis, the scores are split to 'sense' and 'anti-sense'. The promoter scoring method mentioned here was validated by our experimental evolving library: the average score obtained for the 10% already active sequences was 460 ± 229 compared with an average of 85 ± 113 for the rest random sequences that were not active before evolution. Similarly, the average score of the evolved sequences went up from 85 ± 113 before evolution to 230 ± 149 after including the evolved mutations.”

While the proposed metric sounds interesting, I see no evidence one could peg to it. The depletion in motifs might mean many things, and could be confounding in what the authors state as accidental expression. What evidence do the authors have for this? Could there be selection against other features that the authors could be mistakenly be calling against accidental expression?

>> This is a valid concern and we now address this comment by highlighting results from an alternative recoding model and by elaborating more on how our data should be interpreted according to the recoding model used (see lines 251-253, 434-438 and Supplementary Note 6).

Whether or not the wild-type version deviates from an expected composition can be tested after comparing the wild-type scores to the distribution of scores obtained from the versions created according to the expected model. Our results on predicted accidental expression in the coding region should only be interpreted in the light of the recoded models that we used.

Our first recoding model was the standard method that requires the recoded versions of a gene to encode the exact same amino-acid sequence as well as to comply with *E.coli*'s codon bias. Using this recoding model, we observed that the wild-type version is depleted from accidental promoters. These results assure that the depletion observed couldn't be due to a confounding effect that results from the amino-acid sequence or the codon bias. Nonetheless, like the concern raised in this comment, some other cofounding affect might have caused this result.

For this concern specifically, we looked for alternative recoding models that can rule out potential confounding factors other than the ones dismissed by the original recoding model. Since *E.coli*'s promoter motifs are AT-rich we thought that the depletion signal we detected was merely due to fact that when we recode a gene a thousand times (while preserving amino acid sequence and codon bias) many recoded versions will have AT content that deviate from the original AT content of the WT version of the gene. This may lead to more promoter motifs in the recoded versions compared to the WT version, and eventually may seem like the WT version is depleted of promoters, while the truth is that some of the recoded version are enriched for promoters due to the AT content bias.

To rule out this option we redid the analysis using another recoding model, this time each recoded version preserved the exact AT content of the WT original version. The recoding this time was made by reshuffling the original codons of the gene in a thousand different orders. The scores for this recoded method also showed that the WT version is depleted of promoters, indicating that an AT bias is not a confounding factor either. All together we ruled out, amino-acid sequence biases, codon-biases and AT content biases from being a confounding effect that might mislead us to think that the WT coding region is indeed depleted from promotes. In principle there might be other factors that can have a confounding effect, yet we don't have a concrete additional recoding model that to test. Furthermore, our results seem to be supported by other papers that showed general depletion of binding site motifs like Hahn et. al. 2003.

We also further validate that our findings do not stem from potential artifacts of the BPROM software by running an independent scan of the WT genome compared with the recoded versions. This time we simply counted the relative occurrences of all possible six-mer motifs in the genome. This analysis also showed that promoter motifs are depleted from the middle of genes, specifically the -10 motif. Reassuringly, among this group of depleted motifs we also found the Shine-Dalgarno sequence (ribosome binding site).

Here is the modified text:

“Supplementary note 5:

Using an alternative model for recoding of the *E. coli*'s coding sequences

Our first recoding model represent the standard recoding method that requires the recoded versions to encode the exact same amino-acid sequence as well as to comply with the codon bias. This recoding model showed that the wild-type version is depleted from accidental promoters, which means that the depletion observed could not stem from a confounding effect of the amino-acid sequence or the codon bias. Nonetheless, other cofounding affect might have caused the results that showed minimization of accidental expression. Due to this concern, we looked for alternative recoding models that can rule out potential confounding factors other than the ones dismissed by the original recoding model.

Since *E. coli*'s promoter motifs are AT-rich (the two main consensus elements are 10/12 AT, and the additional up elements are either 6/9 or 11/11 AT) we thought that the detected depletion signal was merely due to the fact that when we recode a gene a thousand times (while preserving amino acid sequence and codon bias) some recoded versions will have AT content that deviates from the original AT content of the WT version of the gene. This may lead to more promoter motifs in these recoded versions, and eventually may seem like the WT version is depleted of promoters, while the truth is that some of the recoded version are enriched for promoters due to the AT content bias.

To rule out this option we redid the analysis using another recoding model, this time each recoded version preserved the exact AT content of the WT original version. The recoded versions this time were obtained by reshuffling the original codons of each gene in a thousand different orders. Reassuringly, the results for this recoding method also showed that the WT version is depleted of promoters, indicating that an AT bias is not a confounding factor either (see Supplementary Fig. 4). All together we ruled out, amino-acid sequence biases, codon-biases and AT content biases from being confounding effects that might mislead us to think that the WT coding region is indeed depleted from promoters. In principle there might be other factors that can have a confounding effect, yet at this point we do not have a concrete additional recoding model to test.”

Lines 249-253

“In addition, we performed a “six-mer” analysis - a straightforward unbiased analysis in which we count the number of occurrences for all possible six-mer motifs across the genome. Comparing the differences in six-mer occurrences between the WT genome to the recoded ones revealed which motifs are overrepresented and which are underrepresented in the WT *E. coli* genome (Methods).”

Lines 435-439

“**Six-mer analysis** – Looking for depleted and over represented motifs we counted the occurrences of all six-mers within the coding region of *E. coli*. We compiled a list of all 4096 possible six-mers and counted how many times each six-mer occurs in all WT coding region compared to the 1000 recoded versions. Then, we focused on six-mers that are significantly rare/abundant in WT version compared with their counting in the recoded versions.”

Can this metric be applied to the variety of *E. coli* genomes and transcriptomes, and inspected for its validity in regions where polymorphisms cause variation of "accidental expression" scores and actual transcription readout?

>> We think this is an interesting idea; unfortunately our ability to yield meaningful insights by perusing this approach is low due to some following issues:

Bacterial strains tend to have higher probability for SNPs in intergenic regions compared to coding sequences, because coding regions are generally more conserved than intergenic regions. Therefore, our ability to establish a causal connection between a particular SNP in the coding region (that induce accidental expression) to an effect on gene expression requires that both strains have the exact same sequence upstream to the gene. Furthermore, even minor differences in the growth parameters of these strains (like differences in lag phase, growth rate, media etc.) can result in significant changes in the transcriptome that cannot be distinguished from changes due to induction of accidental expression. Having said that, a recent work by Brophy & Voigt 2016, showed data along this line. Brophy & Voigt created different *E. coli* strains that differ by the type of engineered promoter that was placed to interfere with the expression of a reporter gene, similarly to the “accidental promoters” we were looking for. They measured expression differences with and without the “accidental promoter” and they show that the expression of the reporter gene is indeed hampered. They further suggest such design as a tool to lower gene expression of genes in synthetic biology. Furthermore, when we applied our method on our evolved library of the results reassured the validity of the method. Specifically, our method yields an average score of 85 ± 113 for the random sequences that weren't active before evolution, compared to an average score of 460 ± 229 for the active random sequences. In addition, after evolution the average scores of the mutated sequences went up 85 ± 113 to 230 ± 149 . This data is now part of paragraph discussing our method to assess promoter activity from sequence (lines 429-433).

Here is the modified text:

“The promoter scoring method mentioned here was validated by our experimental evolving library: the average score obtained for the 10% already active sequences was 460 ± 229 compared with an average of 85 ± 113 for the rest random sequences that were not active before evolution. Similarly, the average score of the evolved sequences went up from 85 ± 113 before evolution to 230 ± 149 after including the evolved mutations.”

Can this be explained in other bacterial systems? This is important because the authors often explain their interpretations in evolutionary terms.

>> Promoter recognition machinery that identifies a substantial fraction of random sequences as active raise the question of how cells cope with the associated costs of this promiscuity. We try to tackle this question from an evolutionary perspective and we suggest, as also mentioned by the reviewer, that the genome is not really random, with respect to the probability of finding promoter motifs. In other words, while random sequences contain no information, the genome contains information that reflects the need to be further away from a promoter. We suggest that this notion might be relevant to other microbes too, and indeed a paper by Huerta et al. 2006, showed differential densities of promoter-like sequences between regulatory and nonregulatory regions in most eubacterial genomes.

Another very recent study by de Boer et al. (2017 BioRxiv) may also indicate the need to avoid promoters inside genes. This paper shows that for yeasts too, a fair fraction of random sequences exhibit promoter activity. In their case, they see basal expression from 80% of random sequences. This behavior might result in substantial cost not only due to interference of accidental promoters but also due to 'real' promoters that have to compete with 'accidental promoters' for recruitment of RNAP. In the case of yeast, indications for selection to minimize the costs of promiscuity can be achieved by nucleosome hindering.

To summarize, I think the authors raise interesting points, but the strategies are somewhat incomplete. I think the first part of the manuscript is fine as it is and it provides interesting results, as long as the authors explicitly state that what they are measuring is how short mutational distance of any sequence to a promoter affects its promoter-like effects.

The second part is a bit more problematic, and while interesting as a correlational study, I don't think it has enough support for the vast amount of interpretation the authors go into.

>>We hope that the additional analyses that we have provided will satisfy the referee. See for example new main figure 4, and the corresponding main text paragraph, as well as the six-mer analysis and the corresponding Supplementary Table 2

Here is the paragraph that describes the new figure 4 (lines 205-218):

“Computational assessment of promoter sequence accessibility

Our evolution experiment showed that a single mutation could often produce expression levels similar in magnitude to the expression level produced by the WT lac promoter. To get a numerical perspective on these findings, we assessed the mutational distance that separates random sequences from the canonical promoter of *E. coli*. To this end, we computationally created 30,000 random sequences (the same way the experimental RandSeq1 to 40 were generated) and ran a template of the canonical promoter against their sequences, using a sliding window. Since the importance of each base for promoter activity differs considerably, we weighted each base according to the position-specific matrix of the *E. coli* canonical promoter (Methods). In the same way we also obtained scores for the 556 constitutive promoters³⁹ of *E. coli* and set their median score as the benchmark that qualifies as a promoter (Supplementary Note 3). The results from this analysis showed that the fraction of random sequences that qualify as promoters and the fraction that are one mutation away from a promoter coincide with the fractions observed in our experimental library. Similar fractions were also observed when the benchmark for a promoter was set to capture the core bases of the canonical motifs (TTGnnn and TAnnnT, where 'n' represent any base)(Fig. 4).”

REVIEWERS' COMMENTS:

Reviewer #1 (Remarks to the Author):

The authors have adequately addressed the points I raised in my review. I thank them for their detailed responses to my questions. I have no further comments for them.

Reviewer #2 (Remarks to the Author):

Critique of "Random Sequences Rapidly Evolve into de novo Promoters" by Yona et al, round 2, for Nature Communications

2 March 2018

The authors have addressed my primary concerns. The validation of evolved promoters achieved by mutating the putative -11 adenine in RandSeq1-3 does a lot to bolster the claim that the primary route to promoter evolution is the appearance of -10 and -35 elements. The expanded discussion of this result is helpful in highlighting how nontrivial a finding this is. Also, the new computational analysis in Fig. 4 is valuable for establishing that the authors' findings are reasonably consistent with known biology. A more quantitative treatment of evolution rates would be ideal, but I don't think it's essential.

Overall I think this is now quite a nice piece of work. It will be particularly valuable for the evolution community, I think, since the authors address such a basic (yet previously unaddressed) problem and their results are pretty definitive.

By the way, I want to voice an objection to some of the points that Reviewer #3 makes.

1. The claim that this paper "sheds little light into the matter of how a sequence that is *far* from a motif is able to gain promoter activity" seems incorrect to me. The authors do show other routes to gaining a promoter, e.g., recombination from a different locus. I think it's quite nice, actually, how the authors are able to roughly quantify the rates at which these different evolutionary strategies are used.

2. Also, if it's true that "similar (although not identical) studies have been performed," what are those studies? I don't know of any such studies, nor do the evolutionary biologists I know who have read the author's bioRxiv preprint. I think it's unfair to make this criticism without providing references.

Minor points:

1. There are still a number of typos in the manuscript.

2. Line 153: Perhaps the section heading should be "~10% of random 103bp sequences...". Similarly in line 170. Sequence length is critical in this context.

3. Line 229: Maybe the authors should clarify that a typical ~1kb sequence will contain an accidental promoter *in each direction*.

4. Line 275: The data supporting the claim that essential genes are under stronger pressure against accidental expression are subtle and might reasonably be questioned. The authors might want to

supply some statistics to bolster their claim.

5. Lines 285 and 288: Since the figure showing the results for toxin/antitoxin genes is relegated to the supplement, the authors might consider including some summary statistics in the main text.

6. Line 441: Why not make all data and code available on GitHub? Person-to-person requests for code are usually unsuccessful.

7. Supplementary Note 2: It seems unlikely that TtTtAT serves as a -10 element, since it is missing the -11 A and is implausibly close to the putative -35 element.

Reviewer #3 (Remarks to the Author):

The authors show how many random sequences can provide successful candidates that turn into promoters in a very short period of time, and the implications in general genome biology.

I am impressed by work the authors have undertaken since the first version. I was one of the original reviewers in their previous submission to another journal, and the manuscript has been clearly improved thoroughly. I think the message is much more focused and the analyses have increased their rigor.

I support the manuscript for publication.

We thank the reviewers for their comments. Please see our point-by-point response to the comments raised by reviewer #2 (in blue).

REVIEWERS' COMMENTS:

Reviewer #2 (Remarks to the Author):

Critique of “Random Sequences Rapidly Evolve into de novo Promoters” by Yona et al, round 2, for Nature Communications

2 March 2018

The authors have addressed my primary concerns. The validation of evolved promoters achieved by mutating the putative -11 adenine in RandSeq1-3 does a lot to bolster the claim that the primary route to promoter evolution is the appearance of -10 and -35 elements. The expanded discussion of this result is helpful in highlighting how nontrivial a finding this is. Also, the new computational analysis in Fig. 4 is valuable for establishing that the authors’ findings are reasonably consistent with known biology. A more quantitative treatment of evolution rates would be ideal, but I don’t think it’s essential. >> We added the relevant data on population sizes of the evolved population in terms of cells saturation concentration and the evolving culture volumes. Here is the modified text from the methods section: “Lab evolution was performed on liquid cultures grown on M9+GlyLac by daily dilution of 1:100 into 3ml of fresh medium. Without the ability to utilize lactose the strains that carry random sequences reached a concentration of 3.3×10^8 cells/ml in stationary. Strains that had or acquired the ability to utilize lactose increased their final concentration to $\sim 1.5 \times 10^9$ cells/ml.”

Minor points:

1. There are still a number of typos in the manuscript.
>> We went over the manuscript and corrected typos that we found.
2. Line 153: Perhaps the section heading should be “~10% of random 103bp sequences...”. Similarly in line 170. Sequence length is critical in this context.
>> We added 103bp to the section headings.
3. Line 229: Maybe the authors should clarify that a typical ~1kb sequence will contain an accidental promoter *in each direction*.
>> We now added “both in the ‘sense’ and the ‘anti-sense’ direction”.
4. Line 275: The data supporting the claim that essential genes are under stronger pressure against accidental expression are subtle and might reasonably be questioned. The authors might want to supply some statistics to bolster their claim.
>> We now report a P value < 0.001 of KS test

5. Lines 285 and 288: Since the figure showing the results for toxin/antitoxin genes is relegated to the supplement, the authors might consider including some summary statistics in the main text.

>> We complemented the main text:

“We observed that for 80% of these couples the toxin genes had higher accidental expression compared to their antitoxin counterparts (P=0.016, binomial)”

“We observed that toxins had a much stronger accidental expression in their antisense direction compared to their sense direction (P=0.059, KS test); unlike the antitoxins in which no significant difference was observed (P=0.541, KS test), as largely seen for all genes in the genome”

6. Line 441: Why not make all data and code available on GitHub? Person-to-person requests for code are usually unsuccessful.

>> We now add a link to a GitHub repository in the method section.

7. Supplementary Note 2: It seems unlikely that TtTtAT serves as a -10 element, since it is missing the -11 A and is implausibly close to the putative -35 element.

>> Indeed. We corrected this problem by omitting this promoter prediction.